# WHEN MODELS KNOW WHEN THEY DO NOT KNOW CALIBRATION, CASCADING, AND CLEANING

## ABSTRACT

When a model knows when it does not know, many possibilities emerge. The first question is how to enable a model to recognize that it does not know. A promising approach is to use confidence, computed from the model's internal signals, to reflect its ignorance. Prior work in specific domains has shown that calibration can provide reliable confidence estimates. In this work, we propose a simple, effective, and universal training-free method that applies to both vision and language models, performing model calibration, cascading, and data cleaning to better exploit a model's ability to recognize when it does not know. We first highlight two key empirical observations: higher confidence corresponds to higher accuracy within a single model, and models calibrated on the validation set remain calibrated on a held-out test set. These findings empirically establish the reliability and comparability of calibrated confidence. Building on this, we introduce two applications: 1. Model cascading with calibrated advantage routing and 2. Data cleaning based on mixture of experts. Using the routing signal derived from the comparability of calibrated confidences, we cascade large and small models to improve efficiency with almost no compromise in accuracy, and we further cascade two large state-of-the-art models to achieve performance beyond either model alone. Leveraging multiple experts and their calibrated confidences, we design a simple yet effective data-cleaning method that balances precision and detection rate to identify mislabeled samples in ImageNet and Massive Multitask Language Understanding (MMLU) datasets. Our results demonstrate that enabling models to recognize when they do not know is a practical step toward more efficient, reliable, and trustworthy AI.

## 1 INTRODUCTION

Modern deep neural networks have achieved remarkable success in both vision and language domains. However, models still frequently produce incorrect (26) in regions of ignorance. Confidence, computed from intrinsic model signals, offers a promising path to estimate such ignorance. Prior work (9; 14; 21; 23; 10) has shown in specific domains that model calibration can yield reliable confidence estimates. In this work, we first introduce a unified framework for model calibration that applies to both vision and language models across diverse tasks.

We highlight two core observations about confidence and calibration: (1) For a single model, confidence functions defined as likelihoods from logits exhibit an approximate monotonic relation, where higher-confidence samples tend to have higher accuracy, even without calibration. (2) Confidence calibrated on a validation set remains calibrated on a held-out test set. Based on these observations, we empirically demonstrate the reliability and cross-model comparability of calibrated confidence.

Building on calibrated confidence, we propose two applications that are broadly applicable to both vision and language models. The first application is model cascading. Prior works have introduced routers or gating modules to decide whether an input should be handled by a larger model, an external resource, or a human (22; 19; 12; 3; 24). We propose a training-free cascading method: using calibrated confidence, we construct a reliable routing signal—confidence advantage—that determines whether to invoke a larger model. On tasks spanning image classification (ImageNet-1K (18)), code generation (Mostly Basic Python Problems (MBPP) (2), and BigCodeBench (27)), reasoning and math task(ARC-Challenge (5), GSM8K (6)) , and knowledge-intensive QA (MMLU (11), we

show that this method achieves an effective balance of accuracy and efficiency. We further extend cascading to large–large model pairs, where the combined cascade surpasses the performance of both standalone models. Notably, on ImageNet-1K, cascading two state-of-the-art models achieves accuracy beyond the previous state of the art.

The second application is data cleaning. Previous work (e.g., Confident Learning (15)) has demonstrated that confidence can signal mislabeled data: when a model is highly confident yet disagrees with the given label, the label is likely incorrect. We build upon this idea by using calibrated confidence as a reference signal. Since even calibrated single models may still err, we introduce a mixture-of-experts approach for data cleaning, where calibrated confidence provides a tunable trade-off between precision and detection rate. This method is simple yet effective. We apply it to ImageNet-1K and MMLU test sets, and through large-scale manual verification on ImageNet, we show that it outperforms prior methods in both accuracy and detection power.

In summary, we propose a unified and streamlined framework for calibration, cascading, and cleaning that applies across domains and tasks. Our contributions are threefold: (1) we formalize a unified view of model calibration and empirically validate the reliability and comparability of existing methods; (2) we introduce a confidence-based cascading method, balancing accuracy and efficiency for small–large pairs and achieving superior performance with large–large pairs; and (3) we present a calibration-driven mixture-of-experts data cleaning approach that is simple, effective, and validated through human annotation. Together, these components provide a practical recipe for leveraging when models know when they do not know.

## 2 METHOD

This section introduces our formalization of model calibration, key empirical observations, and two downstream applications. We begin with a unified formulation applicable to both vision and language models, followed by two calibration methods used in our experiments: temperature scaling and Platt scaling. We then present key observations —— higher confidence corresponds to higher accuracy, and calibration learned on validation generalizes to held-out test sets —— which establish that calibrated confidence is both reliable and comparable across models. Building on these findings, we develop two applications: confidence-based model cascading to balance efficiency and accuracy or to combine strong models for improved performance, and agreement-driven data cleaning where confidence naturally balances precision and recall.

### 2.1 ON CALIBRATION OF VISION MODEL AND LANGUAGE MODEL

**Formulation.** Previous work has proposed calibration formulations tailored to specific cases, such as image classification models. Here we provide a unified formulation of model calibration that applies to both vision and language models, accommodates diverse downstream tasks, and supports different calibration methods.

One sentence summary: calibration is making the model's confidence $c_M$ align with the expected score $\mathbb{E}[v]$ from a verifier $v$, where the expectation is estimated as an average within bins.

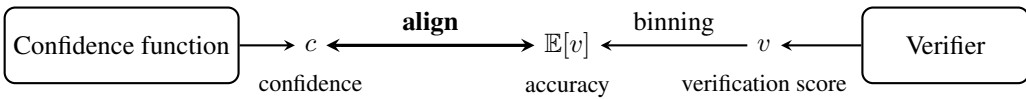

A calibration setup consists of the following components:

- **Model $M$ takes input $X \in \mathcal{X}$ and generates internal states** $z = z_M(X, Y)$, used to sample a prediction $\hat{Y} \in \mathcal{Y}$ and to calculate confidence. For image classification model, $X$ is input image, $Y$ is a class, $z$ is logits over the label set $\mathcal{Y}$; for language model, $X$ is prompt, $Y$ is a possible answer sequence and $z$ is logits over sequence and token space.
- **Verifier $v$ assesses the correctness of an input–answer pair:** Specifically, a verifier $v$ takes a pair $(X, Y)$ as input and outputs a verification score $v(X, Y) \in [0, 1]$. For cases where correctness can be clearly determined, we consider a *binary verifier*, i.e., $v(X, Y) \in \{0, 1\}$ with $v(X, Y) = 0$ (invalid) and $v(X, Y) = 1$ (valid). For example,

in classification tasks (e.g., image classification and multiple-choice benchmarks such as MMLU) where each input $X$ has a corresponding ground-truth label $Y^*$, the verifier is defined as $v(X,Y) = \mathbb{1}^{Y=Y^*}$. For code-generation benchmarks, the verifier checks whether the generated code executes correctly and produces the expected output. A verifier must extract task-relevant information from model outputs, typically in a simplified manner. For example, in MMLU, one may only check whether the first generated token matches the correct label. For code tasks, the verifier needs to extract executable code segments from the output and run them. On average, the binary verification score reduces to *accuracy*.

- **Confidence function** $c(z) \in [0,1]$ **calculates the confidence of an input–output pair based on the model's internal states** $z = z_M(X,Y)$. For classification models, a common choice is $c = \mathrm{softmax}(z/T)$, a temperature-scaled softmax applied to logits, which yields likelihoods over all class labels. For language models, given an output sequence $Y_1 Y_2 \ldots Y_n$, the conditional joint likelihood $\ell(Y \mid X) = \prod_{i=1}^{n} \ell(Y_i \mid X, Y_1, \ldots, Y_{i-1})$ computed via the softmax function can be used as confidence. Alternatively, one may let the language model self-report its confidence, which amounts to an additional sampling step from the model. As noted above, verifier usually evaluates specific subsequences or restricted sub-token spaces. In such cases, the confidence function should also be constrained accordingly. For example, in MMLU only the first token (subsequence) and the four options $\{A, B, C, D\}$ (sub-token space) are relevant. Thus the confidence function should be restricted to this subset and renormalized to preserve its probabilistic interpretation. In what follows, we will also absorb $z_M$ into $c$ and use the shorthand $c_M(X,Y)$.

- **Binning** $\{B_i\}_{i=1}^{N}$ **is a coarse-graining of the data, over which expectations can be estimated.** Binning is essential with binary verifiers, since individual correctness is non-statistical. Accuracy must be considered as an average over equivalence classes (bins). Common binning methods include: **Confidence-based binning:** Partition the interval $[0,1]$ into $N$ equal intervals and assign each sample to a bin according to its confidence value. **Sample-based binning (i.e., histogram equalization):** Sort samples in ascending order of confidence and split them evenly into $N$ bins $\{B_i\}_{i=1}^{N}$. This partitioning prevents samples from concentrating in a few confidence bins, thereby improving robustness. In this work, we adopt sample-based binning.

In general, confidence functions include learnable parameters. The process of optimizing these parameters on a validation set so that the confidence better reflects verifier scores is called **model calibration**. An *ideal calibration* means confidence perfectly reflects accuracy expectation, i.e.,

$$c_M(X,Y) = \mathbb{E}[v(X',Y') \mid (X',Y') \in B_i, \text{ where } (X,Y) \in B_i].$$

In practice, we approximate the expectation by the empirical average, and similarly apply binning average to the confidence, the form of ideal calibration then becomes:

$$\frac{1}{|B_i|} \sum_{(X',Y') \in B_i} c_M(X',Y') = \frac{1}{|B_i|} \sum_{(X',Y') \in B_i} v(X',Y').$$

The left-hand side is the average confidence $\bar{c}_i$ of the bin $B_i$, and the right-hand side is the corresponding average accuracy $\bar{v}_i$. For uncalibrated or non-ideal calibration, we measure the effectiveness of model confidence by the deviation between these two quantities: The Expected Calibration Error (ECE) is defined as:

$$\mathrm{ECE} = \sum_{i=1}^{N} \frac{|B_i|}{n} |\bar{c}_i - \bar{v}_i|,$$

where $n$ denotes the total number of samples, and $|B_i|$ the number of samples in the $i$-th bin.

If the model adopts deterministic sampling such as greedy decoding (so that $\hat{Y}$ is uniquely determined by $X$) and we only care about the confidence of the final prediction, then the above dependence of confidence and accuracy on $(X,Y)$ pairs can be simplified to depend solely on $X$. In what follows, we will also use the shorthand notation $c_M(X)$, which implicitly denotes $c_M(X, \hat{Y})$.

We employ two model calibration methods: *temperature scaling* (for vision tasks) and *Platt scaling* (for language tasks). Both methods involve a few calibration parameters trained on a validation set that is held out from the training and test sets, and both are optimized using the negative log-likelihood (NLL) loss. A notable result is that, despite not directly optimizing ECE, this optimization objective tends to reduce the ECE of the calibrated model, as shown in Fig.1 and Tab.1.

**Temperature Scaling.** Temperature scaling introduces a single learned scalar parameter $T > 0$ applied to the logit vector $z$ over classes to calculate confidence: $c = \text{softmax}(z/T)$. The temperature $T$ is trained using the same objective as in model training, namely the NLL loss over the class labels. Temperature scaling can be used for calibrating multiclass classifiers.

**Platt Scaling.** Platt scaling (16) is a simple parametric calibration method tailored for binary classification. For a model output sequence $Y_1 \ldots Y_n$, we obtain $n$ logit vectors $z_1 \ldots z_n$. Taking their values at the sampled tokens, $\hat{z}_i = z_i^{(Y_i)}$, we use their arithmetic mean $\bar{z} = \frac{1}{n} \sum_i \hat{z}_i$ as a logit estimate for the sequence. Platt scaling then applies a sigmoid function to compute confidence:

$$c = \sigma(a\bar{z} + b),$$

where $a, b \in \mathbb{R}$ are learned parameters and $\sigma$ is the sigmoid function. These parameters are trained by fitting the confidence to the binary verifier's judgment (0 for invalid, 1 for valid) using the NLL loss. Since it is inherently designed for binary outcomes, Platt scaling is particularly suitable for program evaluation tasks, where the result can only be pass or fail.

We next present two key empirical observations that make calibrated confidences reliable:

- **Higher confidence implies higher accuracy.** For the confidence functions defined above, even without calibration, samples with higher confidence tend to have higher accuracy (see Fig.1). The calibration methods described above do not alter the confidence ranking of samples during training, so this approximate monotonic relationship between confidence and accuracy is essential for these methods to be ultimately calibrated.

- **Calibration trained on the validation set generalizes to the test set.** As shown in Fig.1, a confidence function calibrated on the validation set also remains calibrated on the test set. This indicates that confidence reflects the model's underlying estimation of the data distribution, rather than being specialized only to the training set. Hence, calibrated confidence can be expected to remain reliable on in-distribution new data.

The reliability of calibration is also reflected in the quantitative ECE (see Tab.1). Moreover, calibrated confidences across different models become comparable, which enables us to leverage confidence advantage between models to estimate their relative accuracy on specific problems.

## 2.2 CASCADING WITH CALIBRATED ADVANTAGE ROUTING

Building on the comparability of confidences across models, we introduce the notion of *confidence advantage*. Consider two models $M_1$ and $M_2$, define the confidence advantage of $M_1$ compared to $M_2$ in the $i$-th bin $B_i$ as $a_i^c = \bar{c}_i^1 - \bar{c}_i^2$, and the accuracy advantage as $a_i^v = \bar{v}_i^1 - \bar{v}_i^2$, where superscripts 1 and 2 denote the respective model indices.

A natural idea is to use confidence advantage as a routing signal for model cascading. Consider a large model $M_l$ and a small model $M_s$. Both models are calibrated on the validation set, and histogram equalization is applied based on the confidence of the small model. Define the advantage function on the validation set as $a_i^c = \bar{c}_i^l - \bar{c}_i^s$. By the generalization property of the advantage function, for data $x$ outside the validation set, if its confidence under the small model falls into the $i$-th bin, then its expected advantage under the large model will also be close to $a_i^c$. This calibrated advantage function reflects how much the $M_1$ is better than that of $M_2$ on bin $B_i$.

Based on this, we design the following cascading method: consider the bins $\{B_i\}_{i=1}^{N}$, sort them in ascending order of $a_i^c$, and select the top $K$ bins, corresponding to the $K$ bins where the large model shows the smallest advantage over the small model, where $K$ is a hyperparameter balancing accuracy and efficiency. For a new input $x$, we first perform inference with the small model to obtain the output $\hat{Y}$ and confidence $c_s(x)$. If $c_s(x)$ falls into one of the selected $K$ bins, this indicates that the large model has only a marginal advantage on $x$, and thus we adopt the small model's result. Otherwise, if $c_s(x)$ does not belong to these bins, we then invoke the large model for inference and use its output. The pseudocode of this algorithm is provided in Appendix Alg. 1.

While cascading between a large and a small model provides a balance between accuracy and efficiency, we can also cascade two large models to push the performance limit. Our experiments show that such large–large cascades can surpass the performance of either individual model.

## 2.3 DATA CLEANING WITH MIXTURE OF EXPERTS

Even when the verifier contains occasional errors (e.g., mislabeled data in the dataset), calibrated confidence still provides a useful signal about the accuracy of model outputs. This in turn allows us to use model confidence to correct erroneous labels. We demonstrate our approach on data cleaning for the ImageNet and MMLU datasets, which are known to contain mislabeled samples.

**Basic Cleaning.** The basic method is to treat cases where the model's top-1 prediction (the class with the highest confidence) disagrees with the given label as labeling errors. However, this ignores the model's own error probability, even if the model achieves high accuracy. Introducing multiple models as a mixture-of-experts can alleviate this issue. For example, consider two models $M_1$ and $M_2$. The most basic filtering rule is to check whether both $M_1$ and $M_2$ produce top-1 predictions inconsistent with the ground-truth label. If both disagree, the label is likely incorrect. This approach depends on the accuracy of individual models and the independence between experts: the more independent the experts, the better the performance.

**Cleaning with Calibration.** A more refined approach is to incorporate confidence, since models are highly accurate only on samples where they assign high calibrated confidence. A label is then flagged as erroneous when both models' top-1 predictions disagree with the ground-truth label **and** their corresponding confidences fall into the top-$K$ confidence bins, where the hyperparameter $K$ thus controls the trade-off between precision and recall in data cleaning.

**Confident Learning.** As a comparison, previous work has also used calibrated confidence for data cleaning, notably the Confident Learning (CL)(15) framework, in particular its Method 2. Let the given label of an input $X$ be $Y_{\text{label}}(X)$. For each class $Y$, define its class-level average confidence as

$$\bar{c}(Y) = \frac{1}{|\{X' \mid Y_{\text{label}}(X') = Y\}|} \sum_{X':Y_{\text{label}}(X')=Y} c_M(X', Y),$$

where $c_M(X', Y)$ is the calibrated confidence of model $M$ on class $Y$ for input $X'$.

For a new sample $(X, Y_{\text{label}})$, consider the set of candidate classes

$$\mathcal{C}(X) = \{Y \mid c_M(X, Y) > \bar{c}(Y)\}.$$

Among these candidates, take the class $Y^* = \arg\max_{Y \in \mathcal{C}(X)} c_M(X, Y)$. If $Y^* \neq Y_{\text{label}}$, then the label is predicted to be incorrect; otherwise, it is retained as correct. Appendix Alg.2 presents the pseudocodes for the three methods described above.

## 3 EXPERIMENTS

In this section, we first show that calibration is reliable: calibrated confidence is monotonic in accuracy and generalizes from the calibration split to a held-out test split. Building on this, we evaluate cascades in both vision and language. In small–large cascades, our approach assigns to the small model the subset it can solve, achieving a favorable accuracy–small model ratio trade-off; in large–large pairs, our approach exploits complementary strengths to further improve accuracy. Finally, we use cross-model agreement to drive data cleaning and show that higher confidence consistently yields higher cleaning precision. We split the validation set in half, using one part for calibration and the other for testing, and report all results on the test split.

### 3.1 CALIBRATION CAN BE RELIABLE

We first validate the two key conclusions from Section 2.1—that higher confidence correlates with higher accuracy, and that calibration generalizes to the held-out test set. For consistency, we also adopt equal-count binning to partition predictions, reporting bin-wise accuracy and confidence for both validation and test sets.

**Calibration on ImageNet Classification** We first present the results on ImageNet using a single state-of-the-art model, EVA02-Large-448(7). As shown in Figure 1a, the accuracy consistently increases with confidence across all bins, confirming the positive correlation between confidence and accuracy. Moreover, calibration performed on the validation set transfers well to the held-out test

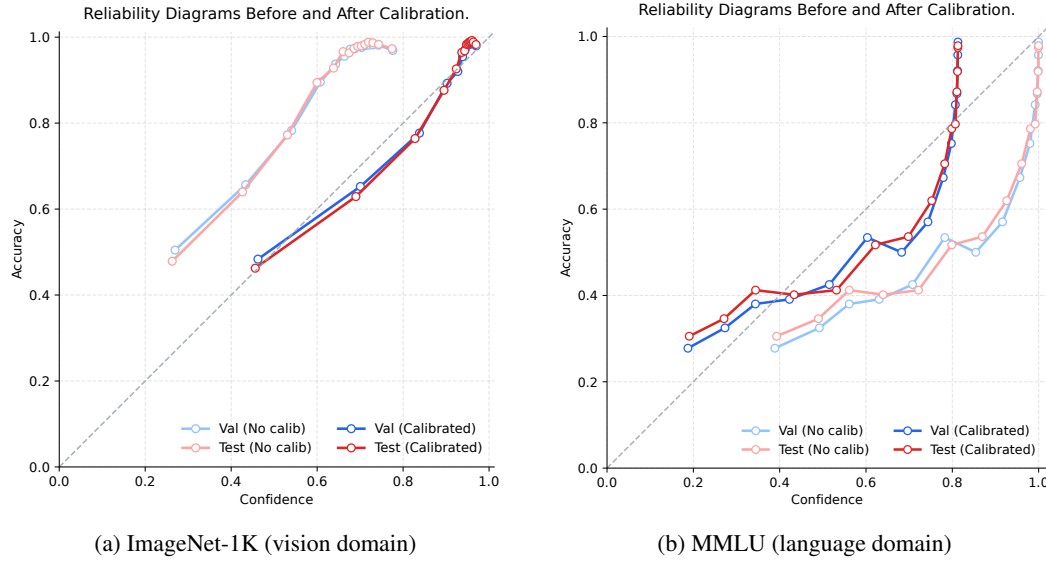

(a) ImageNet-1K (vision domain)    (b) MMLU (language domain)

Figure 1: **Reliability Diagrams Before and After Calibration.** Each panel plots accuracy vs. confidence across 15 equal-count bins. Light curves represent uncalibrated predictions; dark curves represent calibrated predictions. After calibration, confidence becomes more monotonic with accuracy and aligns more closely with the diagonal, indicating improved reliability and generalization from the calibration split to the held-out test split. Left: ImageNet model (`EVA02-Large-448`). Right: MMLU results from a `LLama-8B-Instruct`.

Table 1: **ECE Before and After Calibration on Validation and Test Sets.** Results are reported for both the vision domain (ImageNet) and the language domain (MMLU). Calibration substantially reduces ECE on the validation set, and the improvement generalizes to the held-out test set.

| Dataset | Validation ECE ($\downarrow$) | | Test ECE ($\downarrow$) | |
|---|---|---|---|---|
| | Before | After | Before | After |
| ImageNet | 0.2617 | **0.0271** | 0.2624 | **0.0296** |
| MMLU | 0.1902 | **0.0913** | 0.1830 | **0.0901** |

set, as evidenced by the close alignment between the ECE 1 on validation and test, demonstrating strong generalization of the calibration.

**Calibration on Language Model Generation** We perform the same evaluation procedure on large language models. Specifically, we use calibrated confidence scores from `LLaMA-8B-Instruct`(8) and validate them on the `MMLU` benchmark. As shown in Figure 1b, validation and test accuracies remain closely aligned across confidence bins, confirming that the calibrated confidence scores generalize well even in the language modeling setting.

## 3.2 CASCADE SMALL AND LARGE MODELS

In this section, we use the calibrated confidence advantage as a routing signal for combining models of small and large. Experiments show that the raw confidence advantage already serves as an effective indicator for the portion of data the small model can reliably handle. Calibration further stabilizes this signal, leading to more consistent routing decisions and achieving a better accuracy–small-model ratio trade-off on several datasets.

We apply our proposed method to both image and language models. For evaluation, we consider three baselines:

1. **Random Cascading (RC)** — Data instances are randomly assigned to the small model, and we report the *theoretical* accuracy computed as the average accuracy of the small model rather than running an explicit experiment.

2. **Uncalibrated Confidence-Based Cascading (UC)** — Instances with the highest confidence advantage from the small model are selected without calibration.

3. **Calibrated Confidence-Based Cascading (CC)** — Instances are selected based on calibrated confidence advantage obtained from our calibration procedure.

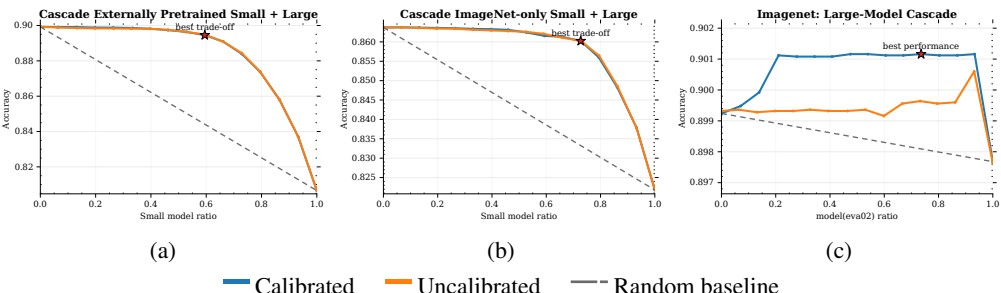

Figure 2: **Cascade Results on ImageNet-1K.** Each panel shows cascade accuracy vs. the routing ratio $p$. Panels (a) and (b) use small–large model pairs (externally pretrained and ImageNet-only, respectively), while panel (c) uses a large–large model pair, showing that calibrated cascading of two strong models further improves accuracy beyond either model alone.

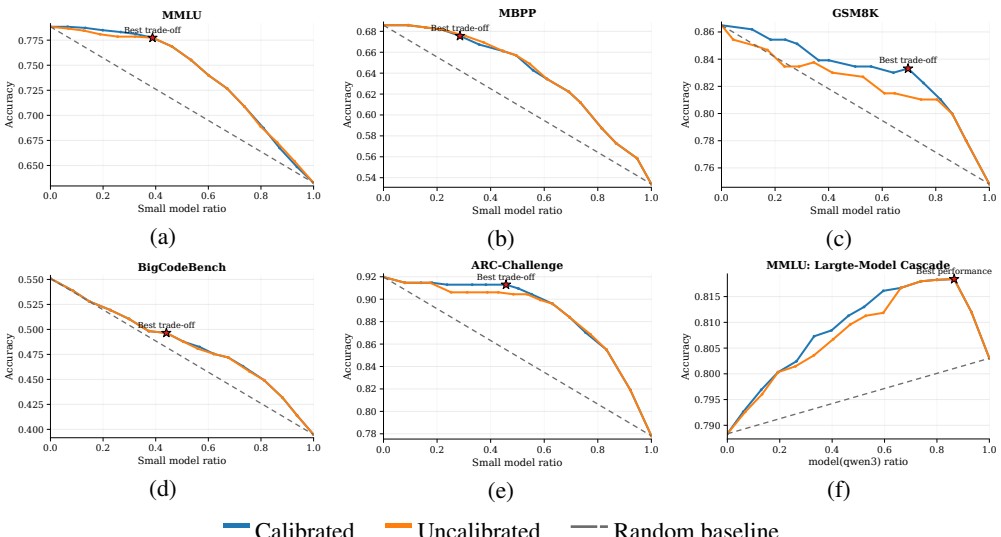

Figure 3: **Cascade Results on Language Models.** Panels (a)–(e) show small–large model cascades on five datasets. Panel (f) shows a large–large cascade, where $p$ denotes the fraction of samples routed to the first large model.

**Cascading Results on ImageNet** In the ImageNet experiments, we evaluate two groups of models as our experimental subjects. The first group consists of a small and a large model that have been trained on data beyond ImageNet. To verify that our approach is not affected by potential data leakage, we also evaluate a second group of models trained solely on ImageNet. As shown in Figure 2, we plot the accuracy curves of UC, CC, and a random-pick baseline as the proportion of small-model predictions increases. Because the small model on ImageNet is already well calibrated, UC and CC yield very similar performance. Nevertheless, both approaches consistently outperform random selection under the same proportion of small-model usage, demonstrating the effectiveness and robustness of our method on image classification models.

**Cascading Results on Language Models** In the language model experiments, we select five diverse and widely adopted datasets: MBPP, BigCodeBench, MMLU, GSM8K, and ARC-Challenge. We

use `LLaMA-70B` as the large model and `LLaMA-8B` as the small model. As show in figure 3, both UC and CC consistently outperform the random-pick baseline across all datasets. Moreover, on `GSM8K` and `MMLU`, CC achieves noticeably better accuracy small-model ratio trade-off compared to UC, highlighting the benefit of using calibrated confidence for routing in the language modeling.

### 3.3 CASCADE LARGE AND LARGE MODELS

In this section, we extend our approach to pairs of large models. We further show that using calibrated confidence as the routing signal enables two models with comparable overall capacity to complement each other by specializing on the data where each performs best, thereby improving overall accuracy. As illustrated in figure 2c, combining two state-of-the-art ImageNet models in this way achieves higher accuracy than either model alone. Similarly, on the `MMLU` dataset, pairing `Qwen-32B` with `LLaMA-70B` yields further performance gains as show in figure 3(f), demonstrating that calibrated confidence not only guides efficient routing but also facilitates effective collaboration between high-capacity models.

Table 2: Cascade accuracy (%) of two EVA-02-Large models on the original and Cleaned half ImageNet validation sets. Model 1: `eva02_large_patch14_448.mim_m38m_ft_in22k_in1k`; Model 2: `eva02_large_patch14_448.mim_in22k_ft_in22k_in1k`.

| Model Combination | Original ImageNet | Cleaned ImageNet |
|---|---|---|
| EVA02-L (M38M) + EVA02-L (IN22k) | 90.116 | 96.266 |

### 3.4 DATA CLEANING

In this section, we demonstrate the effectiveness of leveraging model prediction agreement for data cleaning on both image and language models. We further show that cleaning precision improves as model confidence increases, highlighting a strong positive correlation between confidence and the reliability of the cleaned data.

**ImageNet Cleaning** We use two high-performing ImageNet models (ranked 3rd and 4th[1]as of the time of writing) as complementary experts to flag potential label issues. This avoids using the same models for both cleaning and evaluation, ensuring that the subsequent accuracy measurement remains independent. We compare our method against `CL`, a widely used label-cleaning approach. To assess quality, we randomly sampled 1,000 flagged images for manual verification by trained annotators, which serves as ground truth for estimating precision and detection rate. As shown in Table 3, Our method recovers most mislabeled samples found by `CL` and additionally discovers a much larger set of label errors with high verified precision, demonstrating its stronger ability to improve dataset integrity. We categorize errors into *Wrong Label*, *Multi-Object*, *Similar Class*, *Subclass*, and *Superclass*, with definitions and examples in Appendix B.

Table 3: Comparison between our cleaning method and `CL` on ImageNet. We report the number of samples flagged by each method, the overlap between them, and the accuracy of samples uniquely flagged by each approach.

| Method | Overlap | Flagged Samples | Unique Accuracy (%) |
|---|---|---|---|
| Ours | 1283 | 994 | 0.665 |
| Confident Learning | | 82 | 0.537 |

Furthermore, to validate the conclusion of Section 2.3 — that the accuracy of identified mislabeled samples decreases as confidence decreases — we present the results in Figure 4a. As the detection rate increases (by progressively lowering the confidence threshold), the cleaning precision consistently declines, corroborating our claim. Moreover, across the entire detection range, our method achieves higher cleaning precision than `CL` at the same detection rate, demonstrating the superior effectiveness of our method.

---

[1]`https://github.com/huggingface/pytorch-image-models/blob/main/results/results-imagenet.csv`

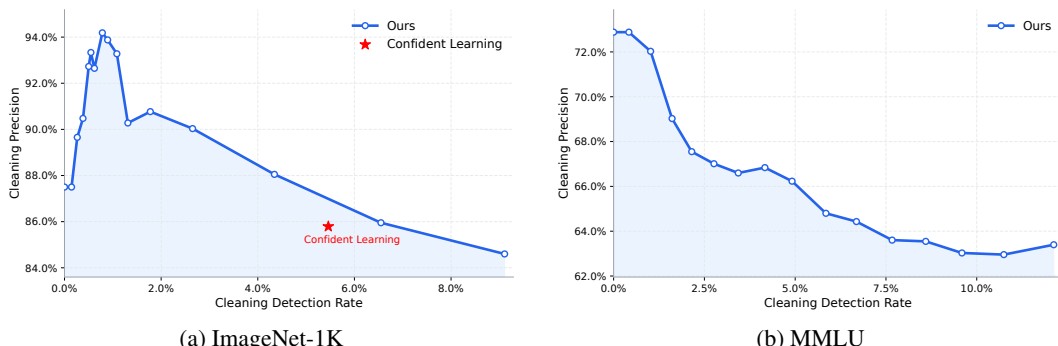

(a) ImageNet-1K  (b) MMLU

Figure 4: **Cleaning Precision vs. Detection Rate.** The horizontal axis (**Cleaning Detection Rate**) is defined as the ratio of flagged samples to the total dataset size. The vertical axis (**Cleaning Precision**) measures the fraction of truly mislabeled samples among the flagged ones, estimated using human verification. Samples are sorted by calibrated confidence in descending order and grouped into 15 equal-count bins; progressively lowering the confidence threshold increases detection rate but typically decreases precision. The red star in panel (a) denotes the operating point of `CL`.

Finally, after completing the data-cleaning process, we apply our cascade method to combine the two state-of-the-art ImageNet models(ranked 1th and 2th). As shown in Table 2, the combined model achieves an impressive accuracy of 96.266%, substantially surpassing the performance of either individual model. This result suggests that, when combined with proper data cleaning, the current state-of-the-art models already approach the upper bound of ImageNet performance, indicating that ImageNet has become a nearly saturated benchmark.

**Language Data Cleaning** Analogous to the image-model setting, we employ `Qwen-32B` (25) and `LLaMA-70B` as expert models to identify potentially mislabeled samples in language datasets. Given the broad coverage of tasks in `MMLU` and the prohibitive cost of large-scale manual verification, we use answers generated by `GPT-4o`(1) in thinking mode as pseudo-labels to estimate precision and detection rate. Applying the same confidence-based filtering procedure as in the vision experiments, we partition the flagged samples into 15 equal-count bins and observe a clear monotonic trend: the precision of flagged samples decreases as bin confidence decreases. This result confirms that calibrated confidence scores remain a robust signal for balancing detection rate and precision in language-model scenarios.

## 4 DISCUSSION AND CONCLUSION

**Relationship with Speculative Decoding** Speculative decoding(13) speeds up generation by letting a smaller model draft candidate tokens and having a larger model verify them, theoretically preserving the large model's output distribution. However, it still runs the large model at every step, incurring additional computation. In contrast, our approach skips large-model evaluation for high-confidence samples, achieving substantial speed gains at the cost of a modest accuracy drop. In short, speculative decoding prioritizes exact fidelity, whereas our method targets a practical accuracy–efficiency trade-off.

**Relationship with Model Routers** Similarly, learned model routers (17; 4; 20) train a separate network to decide which model to use, which introduces extra supervision and complexity. Our approach leverages calibrated confidence as a natural and universal routing signal, requiring no training and generalizing easily across tasks and models.

In summary, this work presents a *universal*, training-free framework for **calibration**, **cascading**, and **cleaning**. We demonstrate that calibrated confidence serves as a natural and general signal for indicating when a model "does not know," enabling it to guide cascading decisions and data cleaning effectively. This simple yet powerful principle allows models to achieve favorable accuracy–efficiency trade-offs, exploit complementary strengths through cascades, and identify mislabeled samples to improve dataset quality. Together, these results highlight how turning model self-awareness into actionable signals can lead to more reliable, efficient, and trustworthy AI systems.

## ETHICS STATEMENT

This study does not involve human participants, sensitive data, or applications with potential for harm. All methods are developed and presented solely for scientific research purposes.

## REPRODUCIBILITY STATEMENT

All experiments in this paper are fully reproducible. We will release our code and datasets upon acceptance to facilitate further research.

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

# Appendix

## A   ANNOTATION AND VERIFICATION PROCEDURE

### A.1   IMAGENET CLEANING VERIFICATION GUIDELINES

**Guidelines for Label-Cleaning Validation (Full Instructions)**

**Columns in the Excel Sheet**

- Image Index
- Image
- Model 1 Prediction
- Model 2 Prediction
- Ground-Truth (GT) Label
- Your Answer

**Important Note.** All provided samples are cases where the predictions from Model 1 and/or Model 2 do *not* match the ground-truth (GT) label. However, the predictions from Model 1 and Model 2 may be identical to each other.

**Your Task.** Determine the correctness of the GT label by referencing the *Image*, the *Model 1 & 2 Predictions*, and the *GT Label*. Based on your evaluation, enter an integer **1–4** in the "Your Answer" cell for that row.

**Classification Categories**

1. **(GT wrong):** The image does not contain the object described by the GT label.

2. **(GT and model correct):** The object in the GT label *and* the object in at least one of the model predictions (Model 1 or 2) are present in the image.

3. **(Only GT correct):** The image *only* contains the object described by the GT label (and does not contain the objects from either of the model predictions).

4. **Unable to determine.**

**Supplementary Instructions**

*Tools.* You may use LLMs (e.g., Gemini, ChatGPT) and search engines to assist your judgment. When using a search engine, adding the keyword "ImageNet" can refine your search. This website can also provide other images from a specific category, e.g., `https://salient-imagenet.cs.umd.edu/explore/`. However, you should not accept an LLM's answer directly. For challenging cases—such as distinguishing between two similar biological species—you must use these tools to learn about the key features and then make your own informed decision.

*Focus.* Your primary goal is to assess the correctness of the GT label. However, it is essential to consider the model predictions, as comparing the different labels is often critical for making an accurate judgment.

*Clarification for Category 1 (GT wrong).* If the image clearly does not contain the object specified by the GT label, you should select Category 1 immediately. You do not need to evaluate whether the model predictions are present in the image.

*Examples for Category 2 (GT and model correct).*

- **Multiple Objects:** The image contains several objects. The model prediction(s) correctly identify one object, while the GT label correctly identifies a different one.

- **Hierarchical/Aspectual Labels:** The model prediction(s) and the GT label describe the image at different levels of specificity or from different perspectives, but both are technically correct (e.g., "off-road vehicle" vs. "four-wheeler"; "laptop" vs. "notebook").

*Handling Lack of Information.*

- If you can clearly determine that the image contains the object from a model prediction, but you lack sufficient information to judge the object in the GT label, choose **Category 1 (GT wrong)**.

- If you can clearly determine that the image contains the object from the GT label, but you lack sufficient information to judge the object(s) in the model prediction(s), choose **Category 3 (Only GT correct)**.

- Choose **Category 4 (Unable to determine)** only when there is insufficient information to determine the presence of *both* the GT label's object and the model predictions' objects.

*Handling Identical Label Names (Rare Case).* In the very rare situation where the GT label and a model prediction are the same name but refer to different actual categories, please select **Category 1 (GT wrong)**.

## A.2 MMLU CLEANING VERIFICATION GUIDELINES

To verify the correctness of samples in the Confmat$_{00}$ set (where both models fail), we reconstruct the original MMLU questions and prompt a GPT model as an independent judge. Each question is presented with its four multiple-choice options, and the model is instructed to answer with *exactly one capital letter (A, B, C, or D)* without explanation. We then compare the model's response with the ground-truth label to estimate the true accuracy of samples in this error region. Temperature is set to $0.0$ to minimize randomness.

## B REPRESENTATIVE EXAMPLES OF IMAGENET LABEL ERRORS

**(1) Wrong Label:** the annotated label does not correspond to any object in the image (e.g., a digital watch labeled as "stopwatch").

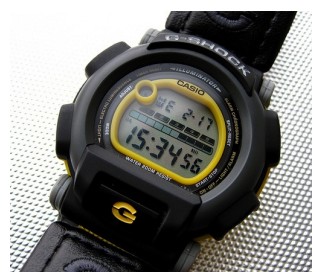 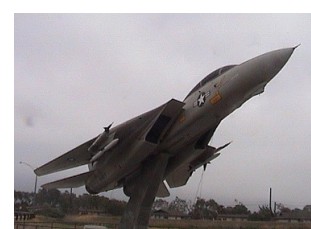 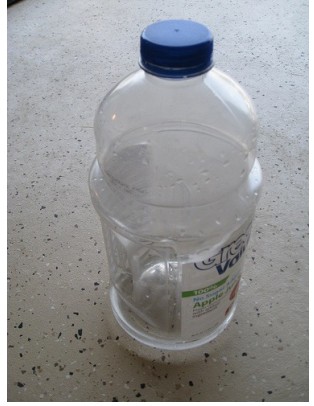

**Label:** stopwatch**True:** digital watch    **Label:** projectile**True:** warplane    **Label:** water jug**True:** water bottle

**(2) Multi-Object:** images containing multiple objects where only one was labeled, leading to incomplete annotations (e.g., "phone" without lamp, "screen" without desk, or "shovel" without the tent).

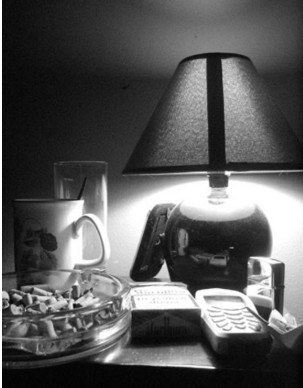 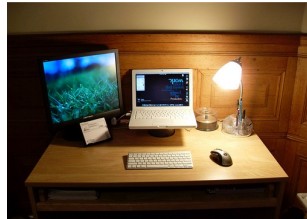 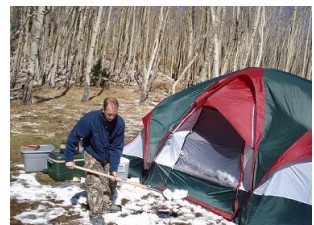

**Label:** phone**True:** phone + lamp    **Label:** screen**True:** screen + desk    **Label:** shovel**True:** shovel + mountain tent

**(3) Similar Class:** cases where the annotated label belongs to a visually similar.

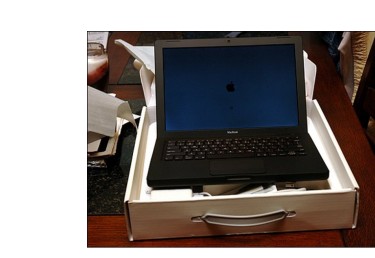 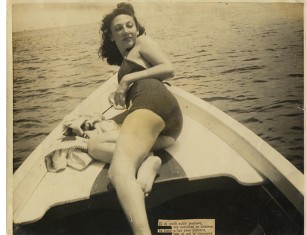 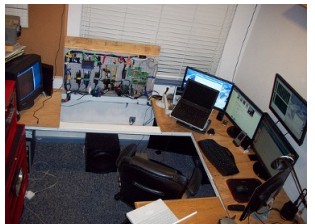

**Label:** desktop**Similar:** notebook  **Label:** maillot**Similar:** maillot  **Label:** screen**Similar:** monitior

**Super/Sub Class:** errors where the annotated label is at a different granularity level from the ground truth. This includes cases where the label is too broad (super-class) or too specific (sub-class) compared to the true category, as well as sibling-class mismatches within the same hierarchy.

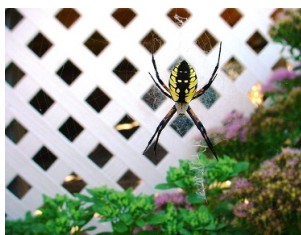 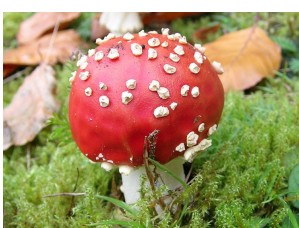 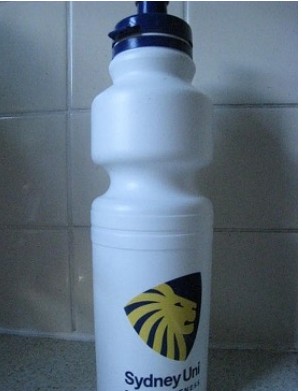

**Label:** garden spider**Sub:** black and gold garden spider  **Label:** mushroom **Sub:** agaric  **Label:** water jug**True:** water bottle

## C  LLM USAGE

LLMs were used only to polish language and improve writing efficiency. All research content is solely by the authors.

# D ALGORITHM PSEUDO-CODES

---

**Algorithm 1:** Model Cascading with Calibrated Advantage Routing

---

**Input:** Large model $M_l$, small model $M_s$; validation set $\mathcal{V}$; number of bins $N$; selection budget $K$ $(1 \leq K \leq N)$.

**Output:** A router that decides whether to use $M_s$ or $M_l$ for a new input $x$.

**Phase I: Calibration and bin statistics on $\mathcal{V}$.**

**Calibrate** $M_l$ and $M_s$ on a held-out validation set $\mathcal{V}$ to obtain calibrated confidence functions $c_l(\cdot)$ and $c_s(\cdot)$.

**foreach** $(X, Y) \in \mathcal{V}$ **do**

    Compute the small-model confidence $c_s(X)$ (or $c_s(X, \hat{Y})$ if label-specific).

Apply histogram equalization on $\{c_s(X) : (X, Y) \in \mathcal{V}\}$ to form $N$ bins $\{b_i\}_{i=1}^{N}$ of (approximately) equal size.

**for** $i \leftarrow 1$ **to** $N$ **do**

    Compute $\bar{c}_i^s := \frac{1}{|b_i|} \sum_{(X,Y) \in b_i} c_s(X)$.

    Compute $\bar{c}_i^l := \frac{1}{|b_i|} \sum_{(X,Y) \in b_i} c_l(X)$.

    Compute the confidence advantage $a_i^c := \bar{c}_i^l - \bar{c}_i^s$.

Sort bins $\{b_1, \ldots, b_N\}$ in ascending order of $a_i^c$ and let $\mathcal{S} \subset \{b_1, \ldots, b_N\}$ be the first $K$ bins (smallest advantages).

**Phase II: Routing on a new input $x$.**

Obtain the small-model prediction $\hat{Y}_s(x)$ and confidence $c_s(x)$.

Find the bin index $i$ such that $c_s(x) \in b_i$.

**if** $b_i \in \mathcal{S}$ **then**

    **return** use $M_s$: output $\hat{Y}_s(x)$.

**else**

    Run the large model to obtain $\hat{Y}_l(x)$; **return** use $M_l$: output $\hat{Y}_l(x)$.

---

---

**Algorithm 2:** Data Cleaning with Calibrated Confidence

---

**Input:** Noisy dataset $\mathcal{D}$, calibrated models $M_1, M_2, \ldots, M_m$, number of bins $N$, threshold $K$.
**Output:** Filtered dataset $\mathcal{D}'$ with corrected labels.
**Method 1: Top-1 Disagreement (Sec. 2.3.1).**
**foreach** $(X, Y_{label}) \in \mathcal{D}$ **do**
    **foreach** *model* $M_j$, $j = 1, \ldots, m$ **do**
        Compute $Y_j^*(X) = \arg\max_Y c_{M_j}(X, Y)$.
    **if** $Y_j^*(X) \neq Y_{label}$ *for all* $j$ **then**
        Mark $Y_{label}$ as erroneous.

**Method 2: High-Confidence Disagreement (Sec. 2.3.2).**
On the validation set, for each model $M_j$, apply histogram equalization to its confidences
$\{c_{M_j}(X)\}$ to form $N$ bins $\{b_i^j\}_{i=1}^N$. Let $\mathcal{B}_{\text{top-}K}^j$ denote the $K$ bins (for model $M_j$) with the
highest average accuracy.
**foreach** $(X, Y_{label}) \in \mathcal{D}$ **do**
    **foreach** *model* $M_j$, $j = 1, \ldots, m$ **do**
        Compute

$$Y_j^*(X) = \arg\max_Y c_{M_j}(X, Y), \quad c_{M_j}(X) = \max_Y c_{M_j}(X, Y).$$

    **if** $Y_j^*(X) \neq Y_{label}$ *for all* $j$ **and** $c_{M_j}(X) \in \mathcal{B}_{top\text{-}K}^j$ *for all* $j$ **then**
        Mark $Y_{label}$ as erroneous.

**Method 3: CLEANLAB-inspired Averaging (Sec. 2.3.3).**
For each class $Y$, compute

$$\bar{c}(Y) = \frac{1}{|\{X' : Y_{\text{label}}(X') = Y\}|} \sum_{X' : Y_{\text{label}}(X') = Y} c_M(X', Y).$$

**foreach** $(X, Y_{label}) \in \mathcal{D}$ **do**
    Define $\mathcal{C}(X) = \{Y : c_M(X, Y) > \bar{c}(Y)\}$.
    Let $Y^*(X) = \arg\max_{Y \in \mathcal{C}(X)} c_M(X, Y)$.
    **if** $Y^*(X) \neq Y_{label}$ **then**
        Mark $Y_{label}$ as erroneous.

---

