# OpenReview forum: "When Models Know When They Do Not Know: Calibration, Cascading and Cleaning"
_ICLR.cc/2026/Conference — Submitted to ICLR 2026_

### Official Review · Reviewer_tASX · 2025-10-25

**Soundness:** 4
**Presentation:** 4
**Contribution:** 4
**Rating:** 8
**Confidence:** 4

**Summary:**

This paper presents a training-free, unified framework leveraging model calibration to enable two capabilities: 1. Model cascading based on calibrated confidence advantage, routing samples between small and large models for efficiency or accuracy gains. 2. Data cleaning via a mixture-of-experts agreement scheme that flags mislabeled examples using calibrated confidences. Experiments span ImageNet, MMLU, GSM8K, MBPP, and BigCodeBench, showing that (i) calibrated confidence monotonically correlates with accuracy and generalizes from validation to test, (ii) cascading can balance cost and accuracy, and (iii) confidence-based cleaning surpasses Confident Learning in precision.

**Strengths:**

I like this paper, it is a complete empirically study, despite the lack of theoicaitcal study.
1. Clarity and simplicity: The framework is easy to understand and reproduce.
2. Cross-domain applicability: Demonstrated on both vision and language tasks.
3. Empirical breadth: Includes multiple benchmarks and models, from EVA-02 to LLaMA/Qwen.
4. Training-free nature: Practical for deployment without retraining routers.
5. Solid comparisons: Includes random baselines and Confident Learning.

**Weaknesses:**

1. Limited novelty: Essentially an aggregation of calibration and routing concepts; lacks new algorithms or theory.
2. No OOD or corrupted-data tests; calibration reliability is only shown in-distribution. The assumption that "better calibration on validation set results in better calibration in test set" may not hold in this setting. Especially in LLM scenario, LLM face domain shift a lot. This should be discussed.
3. Minor improvements: Accuracy and ECE gains are small in some cases. For example the MMLU and imagenet. The explanation that imagenet models are well calibrated so performance are close. But why language task still perform the same? (uncalibrated and calibrated)

**Questions:**

Why don't try some other calibration methods? maybe these models are not well calibrated under such methods.

---

> ### Author Response · Authors · 2025-11-29
> **Author Response to Reviewer tASX (1/3)**
>
> **Q1: “... Essentially an aggregation of calibration and routing concepts; lacks new algorithms or theory.”**
>
> **A1:** Although these ideas may seem familiar, our experiments indicate that even very simple strategies can offer practical value and, somewhat surprisingly, achieve competitive—or in some cases stronger—performance than more complex and computationally intensive approaches. We hope these empirical findings can serve as a useful complement to prior work and provide additional insight to the community. The main contributions of our work are:
>
> 1. **Confidence-based model cascading.** We propose a simple yet effective confidence-based cascading method that **outperforms a widely adopted, more complex, and computationally expensive routing-based approach** [1] (*Table 6*), suggesting that the potential of simple strategies remains underexplored.
>
> 2. **Confidence- and ensemble-based data cleaning.** We introduce a simple and adjustable data-cleaning strategy based on confidence and ensembling, which outperforms widely used prior methods [2] that rely on more complex estimation procedures, and we validate this advantage through large-scale manual annotation.
>
> 3. **Unified framework for calibration.** We provide a unified formal framework for calibration that applies to both language and vision models.
>
> **Q2: “No OOD or corrupted-data tests…”**
>
> **A2:** We added calibration experiments on OOD image datasets (ImageNet-C, ImageNet-R) and an OOD language dataset (MMLU-Adversarial). In addition, we evaluate the performance of the proposed confidence-based model cascading method under these OOD settings. All calibrations are performed on the in-domain validation sets (ImageNet-1K for vision and MMLU for language) while evaluating on out-of-distribution test sets as shown in *Table 1, 2, 3,* and *4*.
>
> *Since figures cannot be included in the response, we report APGR (Average Performance Gap Recovered, adopted from [1]) to measure cascading performance. APGR is defined as the normalized area under the performance–model ratio curve, where the random cascading baseline corresponds to a triangular region with normalized area 0.5.*
>
> - **[Table 1] ECE of Model Calibration on OOD Image Datasets.**
>   - 1a. ImageNet-R
>     | Model                       |   ECE   |
>     |-----------------------------|--------:|
>     | Eva-in22k After Calibration | 0.7711  |
>     | Coat_small After Calibration| 0.6282  |
>
>   - 1b. ImageNet-C Gaussian Noise
>
>     | ECE                        | O-val   | O-test  | C-1     | C-3     | C-5     |
>     |---------------------------|--------:|--------:|--------:|--------:|--------:|
>     | Eva-in22k Before Calibration | 0.2236 | 0.2221 | 0.2261 | 0.2168 | 0.1641 |
>     | Eva-in22k After Calibration  | 0.0253 | 0.0307 | 0.0254 | 0.0284 | 0.0572 |
>     | Coat_small Before Calibration| 0.0440 | 0.0512 | 0.0380 | 0.0357 | 0.0183 |
>     | Coat_small After Calibration | 0.0324 | 0.0396 | 0.0444 | 0.0459 | 0.0584 |
>
>   - ImageNet-C Motion Blur
>
>     | ECE                          |  O-val | O-test |  C-1  |  C-3  |  C-5  |
>     |------------------------------|-------:|-------:|------:|------:|------:|
>     | Eva-in22k Before Calibration | 0.2236 | 0.2221 | 0.2240 | 0.2188 | 0.2014 |
>     | Eva-in22k After Calibration  | 0.0253 | 0.0307 | 0.0250 | 0.0317 | 0.0360 |
>     | Coat_small Before Calibration| 0.0440 | 0.0512 | 0.0367 | 0.0336 | 0.0953 |
>     | Coat_small After Calibration | 0.0324 | 0.0396 | 0.0449 | 0.0727 | 0.1602 |
>
>   - ImageNet-C Fog
>
>     | ECE                          |  O-val | O-test |  C-1  |  C-3  |  C-5  |
>     |------------------------------|-------:|-------:|------:|------:|------:|
>     | Eva-in22k Before Calibration | 0.2236 | 0.2221 | 0.2191 | 0.2228 | 0.2223 |
>     | Eva-in22k After Calibration  | 0.0253 | 0.0307 | 0.0260 | 0.0249 | 0.0326 |
>     | Coat_small Before Calibration| 0.0440 | 0.0512 | 0.2883 | 0.3629 | 0.3763 |
>     | Coat_small After Calibration | 0.0324 | 0.0396 | 0.1806 | 0.2732 | 0.3140 |
>
>
> - **[Table 2] ECE of Model Calibration on OOD Language Datasets — MMLU-Adversarial**
>
>     | Model     | Calibration Status   |  MMLU   | MMLU-Adversarial |
>     |:---------:|:--------------------:|:-------:|:-----------------:|
>     | llama8b   | Before Calibration   | 0.1863  |      0.2496       |
>     | llama8b   | After Calibration    | 0.0185  |      0.0951       |
>     | llama70b  | Before Calibration   | 0.1688  |      0.2004       |
>     | llama70b  | After Calibration    | 0.0147  |      0.0856       |
>
>
> - **[Table 3] APGR of Model Cascading on OOD Image Datasets**
>
>     | Severity          | Gaussian | Motion Blur |   Fog   |
>     |:-----------------:|:--------:|:-----------:|:-------:|
>     | Original ImageNet | 0.7984   |             |         |
>     | 1                 | 0.8158   |   0.8014     | 0.7675  |
>     | 3                 | 0.7588   |   0.7267     | 0.7220  |
>     | 5                 | 0.6364   |   0.5999     | 0.5409  |

---

> ### Author Response · Authors · 2025-11-29
> **Author Response to Reviewer tASX (2/3)**
>
> - **[Table 4] APGR of Model Cascading on OOD Language Datasets**
>
>     | Dataset           |  APGR   |
>     |:-----------------:|:-------:|
>     | MMLU              | 0.7013  |
>     | MMLU-Adversarial  | 0.6301  |
>
> **Observations:**
>
> - As expected, calibration and cascading gradually degrade as the degree of OOD shift increases. Under mild OOD shifts (e.g., ImageNet-C with severity levels 1 and 3, as shown in Table 1 and 3, and language-model evaluation on MMLU-Adversarial), calibrated confidence generalizes reasonably well. Under *moderate* OOD shifts (e.g., ImageNet-C with severity level 5), the method still retains a meaningful degree of generalization, though with a noticeable drop compared to the mild setting. In contrast, severe OOD (e.g., ImageNet-R) leads to a complete failure of calibration.
>
> - Interestingly, while prior work reports that smaller models are often better calibrated in-domain, we observe that after calibration, larger models (such as Eva-in22k) exhibit significantly stronger calibration robustness under OOD settings than smaller models (such as Coat_small). This is likely because Eva-in22k is pretrained on ImageNet-22K, and such large-scale pretraining makes the OOD settings less out-of-domain.
>
> **Q3: “Accuracy and ECE gains are small in some cases…”**
>
> **A3:** For small image models on classification tasks and language models on MMLU, the models are naturally better calibrated, which makes the ECE improvement from calibration appear small in some cases. We therefore added experiments under more severely miscalibrated settings, where substantially larger ECE gains are observed, as shown in *Table 5*.
>
> Regarding the seemingly small accuracy gains before and after calibration in the cascading method, we provide the following explanation:
>
> The proposed confidence-based model cascading method consists of two components:
>
> - **Part 1:** On the validation set of a model, samples are sorted by confidence and histogram equalization is applied to form N bins with equal sample numbers.
>
> - **Part 2:** Post-hoc calibration methods (e.g., temperature scaling or Platt scaling) are applied to obtain calibrated confidence, which is then used to compute the calibrated advantage on each bin.
>
> Importantly, **Part 1 already performs a form of non-parametric confidence estimation using validation data.** The histogram equalization step is closely related to histogram-based calibration [6] and thus already introduces implicit calibration. In Figures 2 and 3, the terms “uncalibrated” and “calibrated” refer only to the confidence *before vs. after post-hoc scaling in Part 2*. Therefore, naming the “uncalibrated” curves as an uncalibrated baseline is inaccurate and misleading.
>
> To avoid confusion, we have added the following experiments to strengthen the evidence and will revise accordingly for better clarity:
>
> - **To demonstrate the impact highlighted in Part 2**, we additionally conduct calibration and cascading experiments on more severely miscalibrated data. As shown in *Table 5*, we re-perform post-hoc calibration on ImageNet-C with severity level 5, where the raw model exhibits poor calibration. After applying temperature scaling, the ECE is reduced to a reasonable range, and the corresponding cascading results are shown in the same table. We observe that as the ECE decreases, the cascading performance consistently improves to varying degrees. This shows that when a model’s calibration error is too large to support reliable cascading decisions, post-hoc calibration becomes necessary.
>
> - **We have added a well-established routing-based method** [1] **and an MoE method** [3] as additional baselines to provide further evidence (Part 1 + Part 2), as shown in *Table 6* and *Table 7*. The results show that the proposed cascading approach in the manuscript significantly outperforms this baseline while maintaining its simplicity. Under the limited rebuttal time, we made our best effort to ensure the faithfulness of all baselines and to avoid performance degradation. We adopted the **original evaluation metrics used in the compared works**: APGR for language tasks, and activated parameter ratio and accuracy for vision tasks.
>
> [Table 5] ECE and APGR of model cascading on more uncalibrated settings. (ImageNet-C, Severity=5)
> | APGR(ECE)             | gaussian_noise | motion_blur |   fog   |
> | :--------------: | :-------------: | :----------: | :-----: |
> | before_calibrated |     0.6364(0.0584)      |    0.5999(0.1602)    | 0.5409(0.3140)  |
> | After_calibrated  |     0.6364(0.0537)      |    0.6122(0.0589)    | 0.6888(0.0285)  |

---

> ### Author Response · Authors · 2025-11-29
> **Author Response to Reviewer tASX (3/3)**
>
> [Table 6] APGR comparison of model cascading methods on language tasks.
> | Method                               |   MMLU   |  GSM8K  |
> | :-----------------------------------: | :------: | :-----: |
> | Random Baseline                      |  0.5000  | 0.5000  |
> | Matrix Factorization (Pretrained)     |  0.5418  | 0.5672  |
> | Matrix Factorization (Trained)        |  0.5267  | 0.6234  |
> | Matrix Factorization (Finetuned)      |  0.5497  | 0.5718  |
> | Calibrated confidence cascading (Histogram Binning, 15 bins)      |  0.7006  | 0.7489  |
> | Calibrated confidence cascading (Temperature Scaling, 15 bins)    |  0.7013  | 0.7593  |
> | Calibrated confidence cascading (Platt Scaling, 15 bins)          |  0.7013  | 0.7496  |
>
> [Table 7] Performance comparison of model cascading methods on language tasks.
> | Method                               | Active Param | Top-1 ACC |
> | :-----------------------------------: | :----------: | :--------: |
> | Single model (DeiT-B)      |   86.57      |   81.98    |
> | Single model (TinyViT)      |   11.00               |   81.53          |
> | MoEE (DeiT-B)       |   58.55      |   80.12    |
> | Calibrated confidence cascading (DeiT-B + TinyViT) (Temperature Scaling, 15 bins K=10) |   39.85   |   90.46   |
> | Calibrated confidence cascading (DeiT-B + TinyViT) (Histogram Binning, Training-Free, 15 bins K=10) |   39.85   |   90.46   |
>
> **Q4: “Why don't try some other calibration methods?”**
>
> **A4**: Good suggestion! We further include a training-free, non-parametric calibration method—histogram binning [6] — as an additional comparison. The following results evaluate three calibration methods: temperature scaling, Platt scaling, and histogram binning, in terms of both ECE and cascading performance. All three methods exhibit comparable performance, which we attribute to the favorable properties of model confidence (approximate monotonicity and generalization).
> [Table 8] ECE of different calibration methods.
> | ECE / Dataset         |  MMLU  | MBPP |  GSM8K  |
> | :------------------- | :----: | :-----------: | :-----: |
> | Histogram Binning    | 0.0239 |    0.0719     | 0.0536  |
> | Temperature Scaling  | 0.0768 |    0.1635     | 0.1054  |
> | Platt Scaling        | 0.0203 |    0.0696     | 0.0731  |
> [Table 9] APGR of model cascading of different calibration methods.
> | AGPR / Dataset        |  MMLU  |  MBPP  |  GSM8K |
> | :------------------- | :----: | :----: | :----: |
> | Histogram Binning    | 0.7006 | 0.6609 | 0.7489 |
> | Temperature Scaling  | 0.7013 | 0.6772 | 0.7593 |
> | Platt Scaling        | 0.7013 | 0.6897 | 0.7496 |
>
> [1] RouteLLM: Learning to Route LLMs with Preference Data. ICLR 2025
>
> [2] Confident Learning: Estimating Uncertainty in Dataset Labels. JAIR
>
> [3] Efficient Data Driven Mixture-of-Expert Extraction from Trained Networks. CVPR 2025
>
> [4] Do Deep Generative Models Know What They Don't Know? ICLR 2019
>
> [5] A Survey on Hallucination in Large Language Models: Principles, Taxonomy, Challenges, and Open Questions. ACM Transactions on Information Systems Jan 2024
>
> [6] Obtaining calibrated probability estimates from decision trees and naive Bayesian classifiers. ICML 2001
>
> [7] Why Normalizing Flows Fail to Detect Out-of-Distribution Data? NeurIPS 2020
>
> [8] WAIC, but Why? Generative Ensembles for Robust Anomaly Detection
>
> [9] Input complexity and out-of-distribution detection with likelihood-based generative models. ICLR 2020

---

### Official Review · Reviewer_xhYV · 2025-10-27

**Soundness:** 2
**Presentation:** 3
**Contribution:** 1
**Rating:** 2
**Confidence:** 4

**Summary:**

This paper proposes training-free frameworks to leverage model confidence for applications in both vision and language domains. The paper formalizes a unified view of model calibration, empirically showing that calibrated confidence is a reliable cross-model predictor of accuracy. They introduce a model cascading method that uses a calibrated advantage signal to route inputs between small and large models for efficiency, or between two large models to achieve stronger performance. They also present a data cleaning method using an MoE approach, where disagreement between highly confident models and the ground-truth is used to identify mislabeled data in datasets like ImageNet and MMLU.

**Strengths:**

The proposed methods for cascading and data cleaning are simple, intuitive, and don't require complex training of auxiliary models. The framework can apply to both computer vision and natural language processing tasks, demonstrating its generalizability.

**Weaknesses:**

1. The evaluation results about the confidence-accuracy relationship in Sec. 3.1 is of minor contribution, since they are common model calibration characteristics.
2. The calibrated and uncalibrated results in Figure 2(a,b) are almost identical, showing insufficient evaluation. The authors are suggested to evaluate on a truly uncalibrated model.
3. The difference of calibration is minimal for benchmarks other than GSM8K and MMLU in Figure 3.
4. Both the confidence cascading and data cleaning frameworks are verified with limited comparisons, e.g., there are confidence routing methods which should also be included for comparison.
5. Minor: typo around Line 389: "as show".

**Questions:**

1. What is the key novelty of the paper? The data cleaning approach is based on the established idea that confident model predictions that disagree with a label indicate a likely error, while the proposed model cascading is very similar to model routing methods.
2. What is the role of calibrated confidence in these frameworks? Currently, the improvement by calibration seems marginal.

---

> ### Author Response · Authors · 2025-11-29
> **Author Response to Reviewer xhYV (1/2)**
>
> **Q1: “The evaluation results … is of minor contribution, since they are common model calibration characteristics.”, “ What is the key novelty of the paper? ...”**
>
> **A1:** In the early sections, we highlight two empirical observations:
> (i) within a given model, higher confidence tends to correspond to higher accuracy; and
> (ii) models calibrated on a validation set often remain well-calibrated on a held-out test set.
>
> We do **not** intend to claim originality or scientific contribution for these observations. Rather, our goal is **pedagogical**: these points are scattered across, or only implicit in, several prior works, and we believe that presenting them clearly helps motivate our methodology. We will revise the text to make this intent explicit and to avoid any potential misunderstanding.
>
> The **main contributions** of our work are:
>
> 1. **Confidence-based model cascading.**
>    We propose a simple yet effective confidence-based cascading method that **outperforms a widely adopted, more complex, and computationally expensive routing-based approach** [1] (Table 3), suggesting that the potential of simple strategies remains underexplored.
>
> 2. **Confidence- and ensemble-based data cleaning.**
>    We introduce a simple and adjustable data-cleaning strategy based on confidence and ensembling, which **outperforms widely used prior methods** [2] that rely on more complex estimation procedures, and we validate this advantage through large-scale manual annotation.
>
> 3. **Unified framework for calibration.**
>    We provide a unified formal framework for calibration that applies to both language and vision models.
>
> We fully acknowledge the contributions of previous studies to these observations and will revise any wording that may cause ambiguity in credit assignment. We will also add the appropriate citations to clearly attribute these findings.
>
> **Q2: “... results in Figure 2(a,b) are almost identical, showing insufficient evaluation… suggested to evaluate on a truly uncalibrated model…”, “The difference of calibration is minimal … in Figure 3”, “What is the role of calibrated confidence in these frameworks? … the improvement by calibration seems marginal.”**
>
> **A2**: The proposed confidence-based model cascading method consists of two components:
>
> - **Part 1:** On the validation set of Model 1, samples are sorted by confidence and histogram equalization is applied to form N bins with equal sample numbers.
> - **Part 2:** Post-hoc calibration methods (e.g., temperature scaling or Platt scaling) are applied to obtain calibrated confidence, which is then used to compute the calibrated advantage on each bin.
>
> Importantly, **Part 1 already performs a form of non-parametric confidence estimation using validation data.**
> The histogram equalization step is closely related to histogram-based calibration [6] and thus already introduces implicit calibration.
> In Figures 2 and 3, the terms **“uncalibrated”** and **“calibrated”** refer only to the confidence **before vs. after post-hoc scaling in Part 2**.
> Therefore, naming the “uncalibrated” curves as an uncalibrated baseline is inaccurate and misleading.
>
> In settings that are naturally better calibrated (e.g., small vision models on classification tasks or language models on certain NLP tasks), the non-parametric estimation in Part 1 is often sufficient.
> In contrast, in more poorly calibrated scenarios (e.g., cascading two large vision models), the calibrated confidence from Part 2 becomes critical.
>
> To avoid confusion, we have added the following experiments to strengthen the evidence and will revise accordingly for better clarity:
>
> - **We additionally conduct calibration and cascading experiments on more severely miscalibrated data.**
>   As shown in *Table 1*, we re-perform post-hoc calibration on ImageNet-C with severity level 5, where the raw model exhibits poor calibration.
>   After applying temperature scaling, the ECE is reduced to a reasonable range, and the corresponding cascading results are shown in the same table.
>   We observe that as the ECE decreases, the cascading performance consistently improves to varying degrees.
>   This shows that when a model’s calibration error is too large to support reliable cascading decisions, post-hoc calibration becomes necessary.
>
> - **We also add a well-established routing-based method [1] and an MoE method [3] as additional baselines** to provide further evidence (Part 1 + Part 2), as shown in *Table 2* and *Table 3*.
>   The results show that the proposed cascading approach significantly outperforms this baseline while maintaining its simplicity.
>   Under the limited rebuttal time, we made our best effort to ensure the faithfulness of all baselines and to avoid performance degradation.
>   We adopted the **original evaluation metrics used in the compared works**:
>   – APGR for language tasks
>   – Activated parameter ratio and accuracy for vision tasks

---

> ### Author Response · Authors · 2025-11-29
> **Author Response to Reviewer xhYV (2/2)**
>
> [Table 1] ECE and APGR of model cascading on more uncalibrated settings. (ImageNet-C, Severity=5)
> | APGR(ECE)             | gaussian_noise | motion_blur |   fog   |
> | :--------------: | :-------------: | :----------: | :-----: |
> | before_calibrated |     0.6364(0.0584)      |    0.5999(0.1602)    | 0.5409(0.3140)  |
> | After_calibrated  |     0.6364(0.0537)      |    0.6122(0.0589)    | 0.6888(0.0285)  |
>
>
> [Table 2] APGR comparison of model cascading methods on language tasks.
> | Method                               |   MMLU   |  GSM8K  |
> | :-----------------------------------: | :------: | :-----: |
> | Random Baseline                      |  0.5000  | 0.5000  |
> | Matrix Factorization (Pretrained)     |  0.5418  | 0.5672  |
> | Matrix Factorization (Trained)        |  0.5267  | 0.6234  |
> | Matrix Factorization (Finetuned)      |  0.5497  | 0.5718  |
> | Calibrated confidence cascading (Histogram Binning[5], Training-Free, 15 bins)      |  0.7006  | 0.7489  |
> | Calibrated confidence cascading (Temperature Scaling, 15 bins)    |  0.7013  | 0.7593  |
> | Calibrated confidence cascading (Platt Scaling, 15 bins)          |  0.7013  | 0.7496  |
>
> [Table 3] Performance comparison of model cascading methods on language tasks.
> | Method                               | Active Param | Top-1 ACC |
> | :-----------------------------------: | :----------: | :--------: |
> | Single model (DeiT-B)      |   86.57      |   81.98    |
> | Single model (TinyViT)      |   11.00               |   81.53          |
> | MoEE (DeiT-B)       |   58.55      |   80.12    |
> | Calibrated confidence cascading (DeiT-B + TinyViT) (Temperature Scaling, 15 bins K=10) |   39.85   |   90.46   |
> | Calibrated confidence cascading (DeiT-B + TinyViT) (Histogram Binning, Training-Free, 15 bins K=10) |   39.85   |   90.46   |
>
> **Q3: “... frameworks are verified with limited comparisons… routing methods which should also be included for comparison.”**
>
> **A3**: As suggested, we have added the following comparisons.
> For **language tasks**, we compare with a widely cited routing-based method [1].
> For **vision tasks**, we compare with an MoE-based approach [3].
> To ensure faithfulness and practicality under rebuttal-time constraints, we adopt the same baselines and follow the original experimental protocols used in the corresponding papers.
>
> For the routing baseline in language models, we make our best effort to faithfully reproduce its performance under three settings:
>
> - **Using the provided pretrained checkpoint**, as the original work claims:
>   *“our routers have learned common characteristics of queries that allow them to distinguish between strong and weak models, generalizing to new models at inference time without additional training”* [1].
>
> - **Fine-tuning the pretrained checkpoint** on the target task dataset.
>
> - **Training the router from scratch** on the target task dataset.
>
> As shown in *Table 2*, across all settings, the results consistently show that the proposed method **significantly outperforms these baselines while maintaining its simplicity**, suggesting that the potential of simple approaches has not yet been fully explored.
>
> For the vision-model baseline, we follow the original paper and use the same **DeiT model** as the large model. In addition, we introduce a much smaller **TinyViT** model as the lightweight model for cascading.
> Notably, we observe that combining two models with individual accuracies in the **81–82%** range yields a cascading accuracy of **90%**, while simultaneously reducing the total number of activated parameters during inference by a substantial margin, as shown in *Table 3*.
>
> **Q4: Minor: typo around Line 389: "as show".**
> **A4**: Thank you for pointing this out. We have corrected the typo.
>
> [1] RouteLLM: Learning to Route LLMs with Preference Data. ICLR 2025
>
> [2] Confident Learning: Estimating Uncertainty in Dataset Labels. JAIR
>
> [3] Efficient Data Driven Mixture-of-Expert Extraction from Trained Networks. CVPR 2025
>
> [4] Obtaining calibrated probability estimates from decision trees and naive Bayesian classifiers. ICML 2001

---

### Official Review · Reviewer_nfCf · 2025-10-27

**Soundness:** 2
**Presentation:** 2
**Contribution:** 1
**Rating:** 2
**Confidence:** 3

**Summary:**

This paper proposes a unified framework for model inference and data cleaning. Based on the rule of confidence calibration, the author presents two primary applications named calibrated model cascading and mixture-of-experts data cleaning. Extensive experiments demonstrate that the proposed method shows superior precision and recall over baselines like confident learning.

**Strengths:**

1.	The proposed framework is easy to deploy. The author presents the proposed method as a universal method that can be applied to both vision and language tasks.

**Weaknesses:**

1.	The key findings lack novelty. The author highlights that “higher confidence corresponds to higher accuracy within a single model, and models calibrated on the validation set remain calibrated on a held-out test set.” However, these findings are basic rules in confidence calibration. The author uses too many sections to illustrate these facts.
2.	The claim for "training-free" is misleading. The author claims that the proposed cascading method is training-free. However, post-hoc calibration still explicitly requires optimizing parameters on a held-out validation set using NLL loss.
3.	The notation is ambiguous. The author uses many $K$ to represent different items, such as "selection budget" in the cascading algorithm (Alg. 1) and the "threshold " in the data cleaning algorithm (Alg. 2).
4.	The author points out that disagreement-based methods depend on the "independence between experts". However, several key experiments use models from the same family, such as LLAMA and EVA-02.
5.	The use of mixture-of-experts (MoE) is confused. The paper repeatedly uses MoE to describe its data cleaning method. However, MoE usually refers to a specific architecture with a dynamic routing mechanism.
6.	Lack of comparison to other multi-model methods. The small-large cascade is motivated by efficiency. However, the paper only compares it to random routing and an uncalibrated variant. I recommend that the author compare their method with the traditional Mixture-of-Experts (MoE) architecture composed of multiple small uncalibrated models. It is unclear how the proposed method compares in its accuracy-efficiency trade-off to MoE.

**Questions:**

1.	Why use different calibrations for vision and language tasks? Temperature scaling is often used in LLM calibration [1-2].

[1] Can LLMs express their uncertainty? NeurlPS, 2023.

[2] Your Pre-trained LLM is Secretly an Unsupervised Confidence Calibrator. NeurlPS, 2025.

---

> ### Author Response · Authors · 2025-11-29
> **Author Response to Reviewer nfCf (1/3)**
>
> **Q1: “The author highlights … However, these findings are basic rules in confidence calibration.”**
>
> **A1:** In the early sections, we highlight two empirical observations:
> (i) within a given model, higher confidence tends to correspond to higher accuracy; and
> (ii) models calibrated on a validation set often remain well-calibrated on a held-out test set.
>
> We do **not** intend to claim originality or scientific contribution for these observations. Rather, our goal is **pedagogical**: these points are scattered across, or only implicit in, several prior works, and we believe that presenting them clearly helps motivate our methodology. We will revise the text to make this intent explicit and to avoid any potential misunderstanding.
>
> The **main contributions** of our work are:
>
> 1. **Confidence-based model cascading.**
>    We propose a simple yet effective confidence-based cascading method that **outperforms a widely adopted, more complex, and computationally expensive routing-based approach** [1] (Table 3), suggesting that the potential of simple strategies remains underexplored.
>
> 2. **Confidence- and ensemble-based data cleaning.**
>    We introduce a simple and adjustable data-cleaning strategy based on confidence and ensembling, which **outperforms widely used prior methods** [2] that rely on more complex estimation procedures, and we validate this advantage through large-scale manual annotation.
>
> 3. **Unified framework for calibration.**
>    We provide a unified formal framework for calibration that applies to both language and vision models.
>
> We fully acknowledge the contributions of previous studies to these observations and will revise any wording that may cause ambiguity in credit assignment. We will also add the appropriate citations to clearly attribute these findings.
>
> **Q2: “The claim for ‘training-free’ is misleading…”**
>
> **A2:** Thank you for pointing this out. We agree that describing our calibration as *training-free* is misleading, as the scaling-based calibration used in the paper indeed relies on learnable parameters. In fact, there exists a truly **parameter-free calibration method—histogram binning** [5], which under our setting achieves comparable or even better performance in both ECE and cascading as shown in *Table 1* and *Table 2*. We add this method as an optional calibration choice and re-emphasize its contribution, as this simple nonparametric approach has been overlooked in some prior work.
>
> Since figures cannot be included in the response, we report **APGR (Average Performance Gap Recovered, adopted from [1])** to measure cascading performance. APGR is defined as the normalized area under the performance–model ratio curve, where the random cascading baseline corresponds to a triangular region with normalized area 0.5.
>
> [Table 1] ECE of different calibration methods.
> | ECE / Dataset         |  MMLU  | MBPP |  GSM8K  |
> | :------------------- | :----: | :-----------: | :-----: |
> | Histogram Binning    | 0.0239 |    0.0719     | 0.0536  |
> | Temperature Scaling  | 0.0768 |    0.1635     | 0.1054  |
> | Platt Scaling        | 0.0203 |    0.0696     | 0.0731  |
>
> [Table 2] APGR of model cascading of different calibration methods.
> | AGPR / Dataset        |  MMLU  |  MBPP  |  GSM8K |
> | :------------------- | :----: | :----: | :----: |
> | Histogram Binning    | 0.7006 | 0.6609 | 0.7489 |
> | Temperature Scaling  | 0.7013 | 0.6772 | 0.7593 |
> | Platt Scaling        | 0.7013 | 0.6897 | 0.7496 |
> Importantly, our framework and applications are not tied to any specific calibration method as shown in Table 2 and 3; the calibration step can be either parametric or non-parametric. We again thank the reviewer for this clarification and acknowledge that our previous use of training-free was inaccurate.
>
> **Q3: “The author uses many K to represent different items…”**
>
> **A3**: Thank you for pointing this out. We revise the notation to avoid confusion. The same symbol K is used in the paper because it serves as a trade-off parameter in both cases.

---

> ### Author Response · Authors · 2025-11-29
> **Author Response to Reviewer nfCf (2/3)**
>
> **Q4: “... disagreement-based methods depend on the "independence between experts". However, several key experiments use models from the same family…”**
>
> **A4**:  ​​For data cleaning in language tasks, we already used models from different families, namely Qwen-32B and LLaMA-70B. For vision data cleaning, we further add experiments using different-family models, Eva-in22k and Caformer_m, for data cleaning on ImageNet-1k. The results show that at a filtering rate of 84.08%, the accuracy reaches 86.75%. In comparison, for the previously used same-family models (Eva-in22k and Eva-m38m), linear interpolation at the same filtering rate yields an accuracy of 85.60%.
>
> Model independence is indeed critical for data cleaning accuracy. Nonetheless, even when using same-family models where independence may be weakened, our method still outperforms prior more complex data cleaning approaches, further demonstrating its effectiveness.
> Moreover, prior work[4] has shown that multiple independently trained models with the same architecture (and thus different parameters) can already improve OOD detection performance. This suggests that model independence is not solely determined by architectural diversity; the parameter randomness induced by stochastic gradient descent also provides a meaningful source of independence.
> For model cascading, we already use different-family models in the vision setting. We further add a different-family language model pair, Qwen-4B + LLaMA-70B, as an additional comparison, as shown in Table 3.
>
> [Table 3]
> |   APGR    |  mmlu  |  gsm8k  |  mbpp  |   arc   |
> | :-------: | :----: | :-----: | :----: | :-----: |
> |Qwen4b+Llama70B | 0.7610 | 0.9697  | 0.9561 | 0.8896  |
> |   Llama-8B+Llama-70B    | 0.7013 | 0.7496  | 0.6897 |    0.7808    |
>
> **Q5: “The use of mixture-of-experts (MoE) is confused…”**
>
> **A5**: We appreciate the reviewer’s suggestion regarding terminology, and we revise the paper to use the more accurate term model ensemble to avoid any potential confusion.
>
> **Q6: “Lack of comparison to other...methods… I recommend that the author compare their method with the traditional Mixture-of-Experts (MoE) architecture composed of multiple small uncalibrated models…”**
>
> **A6**: We add the following comparisons. For language tasks, we compare with a widely cited routing-based method [1]. For vision tasks, we compare with a MoE-based approach [3]. To ensure faithfulness and practicality under the rebuttal time constraints, we adopt the same baselines and experimental protocols as used in the original papers of these methods.
> For the routing baseline in language models, we make our best effort to faithfully reproduce its performance using three settings:
>
> - using the provided pretrained checkpoint (as the original work claims “our routers have learned common characteristics of queries that allow them to distinguish between strong and weak models, generalizing to new models at inference time without additional training”[1]);
> - fine-tuning the pretrained checkpoint on the target task dataset;
> - training the router from scratch on the target task dataset.
>
> As shown in *Table 4*, across all settings, the results consistently show that the proposed method significantly outperforms these baselines while maintaining its simplicity, suggesting that the potential of simple approaches has not yet been fully explored.
>
> **[Table 4] APGR comparison of model cascading methods on language tasks.**
>
> | Method                                           |  MMLU  | GSM8K  |
> |--------------------------------------------------|:------:|:------:|
> | Random Baseline                                  | 0.5000 | 0.5000 |
> | Matrix Factorization (Pretrained)                | 0.5418 | 0.5672 |
> | Matrix Factorization (Trained)                   | 0.5267 | 0.6234 |
> | Matrix Factorization (Finetuned)                 | 0.5497 | 0.5718 |
> | Calibrated confidence cascading (Histogram Binning, 15 bins) | 0.7006 | 0.7489 |
> | Calibrated confidence cascading (Temperature Scaling, 15 bins) | 0.7013 | 0.7593 |
> | Calibrated confidence cascading (Platt Scaling, 15 bins) | 0.7013 | 0.7496 |

---

> ### Author Response · Authors · 2025-11-29
> **Author Response to Reviewer nfCf (3/3)**
>
> For the baseline in vision models, we follow the original paper and use the same DeiT model as the large model. In addition, we introduce a much smaller TinyViT model as the lightweight model for cascading. Notably, we observe that combining two models with individual accuracies in the 81–82% range produces a cascading accuracy of 90%, while simultaneously reducing the total number of activated parameters during inference by a substantial margin, as shown in Table 5.
> [Table 5] Performance comparison of model cascading methods on language tasks.
> | Method                               | Active Param | Top-1 ACC |
> | :-----------------------------------: | :----------: | :--------: |
> | Single model (DeiT-B)      |   86.57      |   81.98    |
> | Single model (TinyViT)      |   11.00               |   81.53          |
> | MoEE (DeiT-B)       |   58.55      |   80.12    |
> | Calibrated confidence cascading (DeiT-B + TinyViT) (Temperature Scaling, 15 bins K=10) |   39.85   |   90.46   |
> | Calibrated confidence cascading (DeiT-B + TinyViT) (Histogram Binning, Training-Free, 15 bins K=10) |   39.85   |   90.46   |
>
> **Q7: “Why use different calibrations for vision and language tasks? Temperature scaling is often used in LLM calibration…”**
>
> **A7**: The proposed framework is universal and compatible with any post-hoc calibration method. We previously adopted Platt scaling for language tasks mainly for practicality, as MMLU and coding tasks provide verifiable binary rewards. In fact, any calibration technique can be applied to both vision and language models within the proposed framework.
> As shown in Table 1 and 2, we evaluate three calibration methods: temperature scaling, Platt scaling, and histogram binning, in terms of both ECE and cascading performance. All three methods exhibit comparable performance, which we attribute to the favorable properties of model confidence (approximate monotonicity and generalization).
>
> [1] RouteLLM: Learning to Route LLMs with Preference Data. ICLR 2025
>
> [2] Confident Learning: Estimating Uncertainty in Dataset Labels. JAIR
>
> [3] Efficient Data Driven Mixture-of-Expert Extraction from Trained Networks. CVPR 2025
>
> [4] WAIC, but Why? Generative Ensembles for Robust Anomaly Detection
>
> [5] Obtaining calibrated probability estimates from decision trees and naive Bayesian classifiers. ICML 2001

---

### Official Review · Reviewer_mxpA · 2025-10-27

**Soundness:** 3
**Presentation:** 2
**Contribution:** 2
**Rating:** 4
**Confidence:** 5

**Summary:**

This submission leverages the calibration property of both vision and language models for two applications: 1) Model cascading and 2) Data cleaning. The main motivations are: higher confidence corresponds to higher accuracy within a single model, and models calibrated on the validation set remain calibrated on a held-out test set. Based on the two observations, this submission can conduct the model cascade based on the calibrated confidence, aiming to improve efficiency. In the meantime, the disagreement of two models can be used to identify mislabeled samples in ImageNet and MMLU. The experiments show validate the usage of a calibrated confidence score works for both applications.

**Strengths:**

+ It is interesting to study the calibrated confidence score for the following applications. The model cascading is a simple and useful example

+ The writing is very clear. The method is well-introduced. The settings of both data clearning and model cascading are clear.

**Weaknesses:**

+ [**Clarification on the two observations**] The main points are: higher confidence corresponds to higher accuracy within a single model, and models calibrated on the validation set remain calibrated on a held-out test set. The first point has been reported in [a], which could be useful to include. The second point should be specified in the context of the in-distribution test set, as the calibration will fail when it comes to out-of-distribution shifts [b].

    [a] Predicting with confidence on unseen distributions

    [b] Can you trust your model's uncertainty? evaluating predictive uncertainty under dataset shift

+ [**No Consideration of Out-of-Distribution (OOD) Settings**] Following up on the above point, the framework is evaluated entirely on in-distribution datasets like ImageNet and MMLU. There’s no evaluation on distribution shifts (e.g., ImageNet-C, ImageNet-R, or adversarial MMLU subsets), where calibration often fails. It would strengthen the paper to either (1) provide evaluations under OOD conditions, or (2) narrow the scope of claims to reflect this limitation more precisely.

+ [**Clarify the usage of Platt scaling (for language)**] Given that Platt scaling is traditionally designed for binary classification, how does it generalize to sequence-level tasks in language models? Is there any connection with the token confidence of the predicted sequence?

+ [**The improvement is somewhat limited**] According to Figures 2 and 3, the improvement of Calibrated over Uncalibrated is limited. Please clarify the advantage. Please clarify what specific advantages calibrated confidence provides in these cases. Furthermore, since the cascade performance depends on the top-K bin selection, how robust is the approach to this hyperparameter?

**Questions:**

- Please clarify that if involving more than two models helps
- If self-reported confidence work in language tasks?
- How sensitive are the cascading and cleaning results to the number of bins or the top-K selection thresholds?

---

> ### Author Response · Authors · 2025-11-29
> **Author Response to Reviewer mxpA (1/4)**
>
> **Q1: “Clarification on the two observations…”**
>
> **A1:** In the early sections, we highlight two empirical observations:
> (i) within a given model, higher confidence tends to correspond to higher accuracy; and
> (ii) models calibrated on a validation set often remain well-calibrated on a held-out test set.
> We do **not** intend to claim originality or scientific contribution for these observations. Rather, our goal is pedagogical: these points are scattered across, or only implicit in, several prior works, and we believe a clearer statement helps motivate our methodology more completely. We will revise the text to make this intent explicit and to avoid any potential misunderstanding.
>
> The main contributions of our work are:
>
> - **Confidence-based model cascading.**
>   We propose a simple yet effective confidence-based cascading method that **outperforms a widely adopted, more complex, and computationally expensive routing-based approach** [1] (*Table 8*), suggesting that the potential of simple strategies remains underexplored.
>
> - **Confidence- and ensemble-based data cleaning.**
>   We introduce a simple and adjustable data-cleaning strategy based on confidence and ensembling, which outperforms widely used prior methods [2] that rely on more complex estimation procedures, and we validate this advantage through large-scale manual annotation.
>
> - **Unified framework for calibration.**
>   We provide a unified formal framework for calibration that applies to both language and vision models.
>
> We fully acknowledge the contributions of previous studies to these observations and will revise any wording that may cause ambiguity in credit assignment. We will also add the appropriate citations to clearly attribute these findings.
>
> **Q2: “No Consideration of Out-of-Distribution (OOD) Settings…”**
>
> **A2:** We add calibration experiments on OOD image datasets (ImageNet-C, ImageNet-R) and an OOD language dataset (MMLU-Adversarial). In addition, we evaluate the performance of the proposed confidence-based model cascading method under these OOD settings. All calibrations are performed on the in-domain validation sets (ImageNet-1K for vision and MMLU for language) while evaluating on out-of-distribution test sets as shown in *Table 1, 2, 3 and 4*.
>
> *Since figures cannot be included in the response, we report APGR (Average Performance Gap Recovered, adopted from [1]) to measure cascading performance. APGR is defined as the normalized area under the performance–model ratio curve, where the random cascading baseline corresponds to a triangular region with normalized area 0.5.*
>
>
> - **[Table 1] ECE of Model Calibration on OOD Image Datasets.**
>   - 1a. ImageNet-R
>     | Model                       |   ECE   |
>     |-----------------------------|--------:|
>     | Eva-in22k After Calibration | 0.7711  |
>     | Coat_small After Calibration| 0.6282  |
>
>   - 1b. ImageNet-C Gaussian Noise
>
>     | ECE                        | O-val   | O-test  | C-1     | C-3     | C-5     |
>     |---------------------------|--------:|--------:|--------:|--------:|--------:|
>     | Eva-in22k Before Calibration | 0.2236 | 0.2221 | 0.2261 | 0.2168 | 0.1641 |
>     | Eva-in22k After Calibration  | 0.0253 | 0.0307 | 0.0254 | 0.0284 | 0.0572 |
>     | Coat_small Before Calibration| 0.0440 | 0.0512 | 0.0380 | 0.0357 | 0.0183 |
>     | Coat_small After Calibration | 0.0324 | 0.0396 | 0.0444 | 0.0459 | 0.0584 |
>
>   - ImageNet-C Motion Blur
>
>     | ECE                          |  O-val | O-test |  C-1  |  C-3  |  C-5  |
>     |------------------------------|-------:|-------:|------:|------:|------:|
>     | Eva-in22k Before Calibration | 0.2236 | 0.2221 | 0.2240 | 0.2188 | 0.2014 |
>     | Eva-in22k After Calibration  | 0.0253 | 0.0307 | 0.0250 | 0.0317 | 0.0360 |
>     | Coat_small Before Calibration| 0.0440 | 0.0512 | 0.0367 | 0.0336 | 0.0953 |
>     | Coat_small After Calibration | 0.0324 | 0.0396 | 0.0449 | 0.0727 | 0.1602 |
>
>   - ImageNet-C Fog
>
>     | ECE                          |  O-val | O-test |  C-1  |  C-3  |  C-5  |
>     |------------------------------|-------:|-------:|------:|------:|------:|
>     | Eva-in22k Before Calibration | 0.2236 | 0.2221 | 0.2191 | 0.2228 | 0.2223 |
>     | Eva-in22k After Calibration  | 0.0253 | 0.0307 | 0.0260 | 0.0249 | 0.0326 |
>     | Coat_small Before Calibration| 0.0440 | 0.0512 | 0.2883 | 0.3629 | 0.3763 |
>     | Coat_small After Calibration | 0.0324 | 0.0396 | 0.1806 | 0.2732 | 0.3140 |
>
>
> - **[Table 2] ECE of Model Calibration on OOD Language Datasets — MMLU-Adversarial**
>
>     | Model     | Calibration Status   |  MMLU   | MMLU-Adversarial |
>     |:---------:|:--------------------:|:-------:|:-----------------:|
>     | llama8b   | Before Calibration   | 0.1863  |      0.2496       |
>     | llama8b   | After Calibration    | 0.0185  |      0.0951       |
>     | llama70b  | Before Calibration   | 0.1688  |      0.2004       |
>     | llama70b  | After Calibration    | 0.0147  |      0.0856       |

---

> ### Author Response · Authors · 2025-11-29
> **Author Response to Reviewer mxpA (2/4)**
>
> - **[Table 3] APGR of Model Cascading on OOD Image Datasets**
>
>     | Severity          | Gaussian | Motion Blur |   Fog   |
>     |:-----------------:|:--------:|:-----------:|:-------:|
>     | Original ImageNet | 0.7984   |             |         |
>     | 1                 | 0.8158   |   0.8014     | 0.7675  |
>     | 3                 | 0.7588   |   0.7267     | 0.7220  |
>     | 5                 | 0.6364   |   0.5999     | 0.5409  |
>
>
> - **[Table 4] APGR of Model Cascading on OOD Language Datasets**
>
>     | Dataset           |  APGR   |
>     |:-----------------:|:-------:|
>     | MMLU              | 0.7013  |
>     | MMLU-Adversarial  | 0.6301  |
>
> **Observations:**
>
> - As expected, calibration and cascading gradually degrades as the degree of OOD shift increases. Under mild OOD shifts (e.g., ImageNet-C with severity levels 1 and 3, as shown in Table 1 and 3, and language-model evaluation on MMLU-Adversarial), calibrated confidence generalizes reasonably well. Under **moderate** OOD shifts (e.g., ImageNet-C with severity level 5), the method still retains a meaningful degree of generalization, though with a noticeable drop compared to the mild setting. While severe OOD (e.g., ImageNet-R) leads to a complete failure of calibration.
>
> - Interestingly, while prior work reports that smaller models are often better calibrated in-domain, we observe that after calibration, larger models (such as Eva-in22k) exhibit significantly stronger calibration robustness under OOD settings than smaller models (such as Coat_small). But Eva-in22k has been pretrained on ImageNet-22K and such larger-scale pretraining makes the OOD settings less out-of-domain.
>
> One can introduce additional probability density estimation to mitigate overconfidence under OOD settings. However, even generative models that explicitly estimate density still face substantial risks of misclassification in OOD detection. For example, flow-based models may assign higher likelihood to OOD samples [4][7][8][9], and large language models can exhibit hallucinations [5]. Precise OOD handling requires precise generalization, which we believe is one of the core challenges of this field. Despite many existing improvements (such as [7][8][9]), reliable OOD detection remains an open problem.
>
> **Q3: Clarify the usage of Platt scaling (for language)**
>
> **A3:** The proposed framework is **universal** and compatible with **any post-hoc calibration method**.
> We previously adopted Platt scaling for language tasks mainly for practicality, as MMLU and coding tasks provide verifiable binary rewards.
> In fact, **any calibration technique can be applied to both vision and language models within the proposed framework**.
>
> To make this point clearer, we further include a **training-free, non-parametric calibration method—histogram binning [6]** — as an additional comparison.
> The following results evaluate three calibration methods: temperature scaling, Platt scaling, and histogram binning, in terms of both ECE and cascading performance as shown in *Table 5* and *Table 6*.
> All three methods exhibit **comparable performance**, which we attribute to the favorable properties of model confidence (approximate monotonicity and generalization).
>
> **[Table 5] ECE of different calibration methods.**
>
> | ECE / Dataset      |  MMLU   |  MBPP   | GSM8K  |
> | :----------------- | ------: | ------: | -----: |
> | Histogram Binning  | 0.0239  | 0.0719  | 0.0536 |
> | Temperature Scaling| 0.0768  | 0.1635  | 0.1054 |
> | Platt Scaling      | 0.0203  | 0.0696  | 0.0731 |
>
> **[Table 6] APGR of model cascading of different calibration methods.**
> | AGPR / Dataset        |  MMLU  |  MBPP  |  GSM8K |
> | :------------------- | :----: | :----: | :----: |
> | Histogram Binning    | 0.7006 | 0.6609 | 0.7489 |
> | Temperature Scaling  | 0.7013 | 0.6772 | 0.7593 |
> | Platt Scaling        | 0.7013 | 0.6897 | 0.7496 |

---

> ### Author Response · Authors · 2025-11-29
> **Author Response to Reviewer mxpA (3/4)**
>
> **Q4: “According to Figures 2 and 3, the improvement of Calibrated over Uncalibrated is limited… the cascade performance depends on the top-K bin selection, how robust is the approach to this hyperparameter?”**
>
> **A4:** The proposed confidence-based model cascading method consists of two components:
>
> - **Part 1:** On the validation set of Model 1, samples are sorted by confidence and histogram equalization is applied to form N bins with equal sample numbers.
>
> - **Part 2:** Post-hoc calibration methods (e.g., temperature scaling or Platt scaling) are applied to obtain calibrated confidence, which is then used to compute the calibrated advantage on each bin.
>
> Importantly, **Part 1 already performs a form of non-parametric confidence estimation using validation data.**
> The histogram equalization step is closely related to histogram-based calibration [6] and thus already introduces implicit calibration.
> In Figures 2 and 3, the terms “uncalibrated” and “calibrated” refer only to the confidence **before vs. after post-hoc scaling in Part 2**.
> Therefore, naming the “uncalibrated” curves as an uncalibrated baseline is inaccurate and misleading.
>
> In settings that are naturally better calibrated (e.g., small vision models on classification tasks or language models on certain NLP tasks), the non-parametric estimation in Part 1 is often sufficient.
> In contrast, in more poorly calibrated scenarios (e.g., cascading two large vision models), the calibrated confidence from Part 2 becomes critical.
>
> To avoid confusion, we have added the following experiments to strengthen the evidence and will revise accordingly for better clarity:
>
> - To demonstrate the impact highlighted in Part 2, we additionally conduct calibration and cascading experiments on more severely miscalibrated data. As shown in *Table 7*, we re-perform post-hoc calibration on ImageNet-C with severity level 5, where the raw model exhibits poor calibration. After applying temperature scaling, the ECE is reduced to a reasonable range, and the corresponding cascading results are shown in the same table. We observe that as the ECE decreases, the cascading performance consistently improves to varying degrees. This shows that when a model’s calibration error is too large to support reliable cascading decisions, post-hoc calibration becomes necessary.
>
> - We have added a well-established routing-based method Matrix Factorization [1] and an MoE method MoEE [3] as additional baselines to further validate the effectiveness of our full method (Part 1 + Part 2), as shown in *Table 8* and *Table 9*. The results show that the proposed cascading approach in the manuscript **significantly outperforms these baselines while maintaining its simplicity**. Under the limited rebuttal time, we made our best effort to ensure the faithfulness of all baselines and to avoid performance degradation. We adopted the **original evaluation metrics used in the compared works**: APGR for language tasks, and **activated parameter ratio and accuracy** for vision tasks.
>
> [Table 7] ECE and APGR of model cascading on more uncalibrated settings. (ImageNet-C, Severity=5)
> | APGR(ECE)             | gaussian_noise | motion_blur |   fog   |
> | :--------------: | :-------------: | :----------: | :-----: |
> | before_calibrated |     0.6364(0.0584)      |    0.5999(0.1602)    | 0.5409(0.3140)  |
> | After_calibrated  |     0.6364(0.0537)      |    0.6122(0.0589)    | 0.6888(0.0285)  |
>
> [Table 8] APGR comparison of model cascading methods on language tasks.
> | Method                               |   MMLU   |  GSM8K  |
> | :-----------------------------------: | :------: | :-----: |
> | Random Baseline                      |  0.5000  | 0.5000  |
> | Matrix Factorization (Pretrained)     |  0.5418  | 0.5672  |
> | Matrix Factorization (Trained)        |  0.5267  | 0.6234  |
> | Matrix Factorization (Finetuned)      |  0.5497  | 0.5718  |
> | Calibrated confidence cascading (Histogram Binning, 15 bins)      |  0.7006  | 0.7489  |
> | Calibrated confidence cascading (Temperature Scaling, 15 bins)    |  0.7013  | 0.7593  |
> | Calibrated confidence cascading (Platt Scaling, 15 bins)          |  0.7013  | 0.7496  |
>
> [Table 9] Performance comparison of model cascading methods on language tasks.
> | Method                               | Active Param | Top-1 ACC |
> | :-----------------------------------: | :----------: | :--------: |
> | Single model (DeiT-B)      |   86.57      |   81.98    |
> | Single model (TinyViT)      |   11.00               |   81.53          |
> | MoEE (DeiT-B)       |   58.55      |   80.12    |
> | Calibrated confidence cascading (DeiT-B + TinyViT) (Temperature Scaling, 15 bins K=10) |   39.85   |   90.46   |
> | Calibrated confidence cascading (DeiT-B + TinyViT) (Histogram Binning, Training-Free, 15 bins K=10) |   39.85   |   90.46   |

---

> ### Author Response · Authors · 2025-11-29
> **Author Response to Reviewer mxpA (4/4)**
>
> Regarding the parameter K, the curves are obtained by varying K, which controls the trade-off between the proportion of small-model inference and overall accuracy. In practice, K can be selected using the validation curves to achieve a desired ratio–accuracy balance. For all values of K, our cascading method consistently and significantly outperforms the baseline, demonstrating its robustness.
>
> **Q5: “Please clarify that if involving more than two models helps.”**
>
> **A5:** We were not sure whether the suggestion referred to involving more than two models in **model cascading** or in **data cleaning**, so we conducted experiments for both settings.
>
> - **Model cascading (more than two models).**
>   Numerous extensions beyond a two-model configuration are possible. As an initial exploration, we evaluate a simple three-model variant: a small model (Coat_small) paired with two large models (Eva-in22k and Eva-m38m) of comparable scale and accuracy.
>   The small model first processes inputs following the standard cascade; unresolved samples are then routed to one of the two large models based on confidence—the batch is sent to the large model with the higher predicted confidence.
>   With this three-model cascading scheme, we achieve **89.1% accuracy**, while the small model (only 7% of the parameters of the large model) handles **over 60%** of the samples. This performance is comparable to Eva-in22k alone (**89.7%**), showing that multi-model cascading can retain strong accuracy while improving efficiency.
>
> - **Data cleaning (more than two models).**
>   Introducing more than three models tends to **reduce detection rate** while **slightly increasing filtering accuracy**. This trade-off can largely be achieved by the confidence-based filtering mechanism described in our method (see Figure 4 in the main paper), suggesting that adding extra models provides limited additional benefit.
>   Empirically, using **three models** yields a filtering rate of **7.9%** with **86.4%** accuracy.
>   For comparison, using **two models** with linear interpolation achieves **85.3%** accuracy at the same filtering rate, indicating only a **modest improvement** when more models are added.
>
> **Q6: “Does self-reported confidence work in language tasks?”**
>
> **A6:** This is an interesting question but is beyond the scope of the present study.
>
> - **General applicability.**
>   Our framework is model-agnostic and can naturally incorporate self-reported confidence for language tasks, as it only requires a confidence score (regardless of whether it is model-generated or self-reported).
>
> - **Scope considerations.**
>   To ensure a clean and controlled comparison with prior work, we intentionally focus on probability-based confidence and do not explore self-reported confidence in this manuscript.
>
> We agree that this is a promising direction and plan to investigate it in future work.
>
> **Q7: “How sensitive are the cascading and cleaning results to the number of bins or the top-K selection thresholds?”**
>
> **A7:** We added experiments analyzing how the hyperparameter **number of bins** influences the cascading performance. The results show that the method remains **stable across a wide range of bin choices (5–100)**, as summarized in Table 10.
>
> **[Table 10] Effect of number of bins on model cascading.**
>
> | Dataset | 5      | 15     | 50     | 100    |
> |--------|--------|--------|--------|--------|
> | mmlu   | 0.6917 | 0.7013 | 0.7044 | 0.7035 |
> | gsm8k  | 0.7378 | 0.7496 | 0.7547 | 0.7551 |
>
> Regarding the parameter **K**, as discussed in **A4**, the cascading curves are generated by varying *K*, which determines the trade-off between small-model usage and overall accuracy. In practice, *K* can be selected using validation curves to meet a desired ratio–accuracy balance.
>
> **Across all values of K**, the proposed cascading method **consistently and significantly outperforms the baseline**, demonstrating its **robustness**.
>
> [1] RouteLLM: Learning to Route LLMs with Preference Data. ICLR 2025
>
> [2] Confident Learning: Estimating Uncertainty in Dataset Labels. JAIR
>
> [3] Efficient Data Driven Mixture-of-Expert Extraction from Trained Networks. CVPR 2025
>
> [4] Do Deep Generative Models Know What They Don't Know? ICLR 2019
>
> [5] A Survey on Hallucination in Large Language Models: Principles, Taxonomy, Challenges, and Open Questions. ACM
> Transactions on Information Systems Jan 2024
>
> [6] Obtaining calibrated probability estimates from decision trees and naive Bayesian classifiers. ICML 2001
>
> [7] Why Normalizing Flows Fail to Detect Out-of-Distribution Data? NeurIPS 2020
>
> [8] WAIC, but Why? Generative Ensembles for Robust Anomaly Detection
>
> [9] Input complexity and out-of-distribution detection with likelihood-based generative models. ICLR 2020

---

### Author Response · Authors · 2025-11-28
**General reply (1/5)**

We thank our reviewers for their encouraging comments, helpful suggestions, and insightful questions. **All reviewers acknowledge the generality and simplicity of the proposed confidence-based model cascading and data cleaning methods.** R1(mxpA) says “... model cascading is a simple and useful example.”; R2 (nfCf) says “The proposed framework is easy to deploy... as a universal method that can be applied to both vision and language tasks.”; R3(xhYV) says “..proposed methods .. are simple, intuitive, and don't require complex training of auxiliary models… The framework can apply to both computer vision and natural language processing tasks..”; R4 tASX says, “framework is easy to understand and reproduce… Cross-domain applicability...” .

The questions and suggestions help us revise the paper to deliver key messages better. All the reviewers suggest we highlight the contributions of this work, which we address first in this general reply. R1 and R42 requested additional discussion on out-of-distribution (OOD) datasets, while R2 and R3 asked for comparisons with other relevant methods. R1, R3 and R4 asked for clarification on why calibration in the model cascading method appears to provide only marginal improvement. In addition, R1, R2 and R4 raised questions regarding the specific calibration methods used in our paper. We address these shared concerns explicitly in the General Comments section.

R1 = Reviewer mxpA, R2 = Reviewer nfCf, R3 = Reviewer xhYV, R4 = Reviewer tASX.

Since figures cannot be included in the response, we report APGR (Average Performance Gap Recovered, adopted from [1]) to measure cascading performance. APGR is defined as the normalized area under the performance–model ratio curve, where the random cascading baseline corresponds to a triangular region with normalized area 0.5.

###  1. On Contribution and Novelty
In the early sections, we highlight two empirical observations: (i) within a given model, higher confidence tends to correspond to higher accuracy; and (ii) models calibrated on a validation set often remain well-calibrated on a held-out test set. We do not intend to claim originality or scientific contribution for these observations. Rather, our goal is pedagogical: these points are scattered across, or only implicit in, several prior works, and we believe a clearer statement helps motivate our methodology more completely. We will revise the text to make this intent explicit and to avoid any potential misunderstanding, and add the relevant citations accordingly.
The main contributions of our work are:

- **Confidence-based model cascading.**  We propose a simple yet effective confidence-based cascading method that **outperforms a widely adopted, more complex, and computationally expensive routing-based approach [1] (Table 5), suggesting that the potential of simple strategies remains underexplored.**

- **Confidence- and ensemble-based data cleaning.**  We introduce a concise data-cleaning strategy based on confidence and ensembling, which outperforms widely used prior methods [2] that rely on more complex estimation procedures. We further validate this advantage through large-scale manual annotation.

- **Unified framework for calibration.**  We provide a unified formal framework for calibration that applies to both language and vision models, offering conceptual clarity and bridging methodological gaps across modalities.

We fully acknowledge the contributions of previous studies to these observations and will revise any wording that may cause ambiguity in credit assignment. We will also add the appropriate citations to clearly attribute these findings.

---

> ### Author Response · Authors · 2025-11-28
> **General reply (2/5)**
>
> ###  2. Out-of-Distribution(OOD) Settings.
> R1 and R4 note the current evaluation is limited to in-domain datasets, and request an analysis of model calibration under OOD settings.
>
> This is a fantastic suggestion and it is important to include such discussion. We have added calibration experiments on OOD image datasets (ImageNet-C, ImageNet-R) and an OOD language dataset (MMLU-Adversarial). In addition, we evaluate the performance of the proposed confidence-based model cascading method under these OOD settings. All calibrations are performed on the in-domain validation sets (ImageNet-1K for vision and MMLU for language).
>
> - **[Table 1] ECE of Model Calibration on OOD Image Datasets.**
>   - 1a. ImageNet-R
>     | Model                       |   ECE   |
>     |-----------------------------|--------:|
>     | Eva-in22k After Calibration | 0.7711  |
>     | Coat_small After Calibration| 0.6282  |
>
>   - 1b. ImageNet-C Gaussian Noise
>
>     | ECE                        | O-val   | O-test  | C-1     | C-3     | C-5     |
>     |---------------------------|--------:|--------:|--------:|--------:|--------:|
>     | Eva-in22k Before Calibration | 0.2236 | 0.2221 | 0.2261 | 0.2168 | 0.1641 |
>     | Eva-in22k After Calibration  | 0.0253 | 0.0307 | 0.0254 | 0.0284 | 0.0572 |
>     | Coat_small Before Calibration| 0.0440 | 0.0512 | 0.0380 | 0.0357 | 0.0183 |
>     | Coat_small After Calibration | 0.0324 | 0.0396 | 0.0444 | 0.0459 | 0.0584 |
>
>   - ImageNet-C Motion Blur
>
>     | ECE                          |  O-val | O-test |  C-1  |  C-3  |  C-5  |
>     |------------------------------|-------:|-------:|------:|------:|------:|
>     | Eva-in22k Before Calibration | 0.2236 | 0.2221 | 0.2240 | 0.2188 | 0.2014 |
>     | Eva-in22k After Calibration  | 0.0253 | 0.0307 | 0.0250 | 0.0317 | 0.0360 |
>     | Coat_small Before Calibration| 0.0440 | 0.0512 | 0.0367 | 0.0336 | 0.0953 |
>     | Coat_small After Calibration | 0.0324 | 0.0396 | 0.0449 | 0.0727 | 0.1602 |
>
>   - ImageNet-C Fog
>
>     | ECE                          |  O-val | O-test |  C-1  |  C-3  |  C-5  |
>     |------------------------------|-------:|-------:|------:|------:|------:|
>     | Eva-in22k Before Calibration | 0.2236 | 0.2221 | 0.2191 | 0.2228 | 0.2223 |
>     | Eva-in22k After Calibration  | 0.0253 | 0.0307 | 0.0260 | 0.0249 | 0.0326 |
>     | Coat_small Before Calibration| 0.0440 | 0.0512 | 0.2883 | 0.3629 | 0.3763 |
>     | Coat_small After Calibration | 0.0324 | 0.0396 | 0.1806 | 0.2732 | 0.3140 |
>
>
> - **[Table 2] ECE of Model Calibration on OOD Language Datasets — MMLU-Adversarial**
>
>     | Model     | Calibration Status   |  MMLU   | MMLU-Adversarial |
>     |:---------:|:--------------------:|:-------:|:-----------------:|
>     | llama8b   | Before Calibration   | 0.1863  |      0.2496       |
>     | llama8b   | After Calibration    | 0.0185  |      0.0951       |
>     | llama70b  | Before Calibration   | 0.1688  |      0.2004       |
>     | llama70b  | After Calibration    | 0.0147  |      0.0856       |
>
>
> - **[Table 3] APGR of Model Cascading on OOD Image Datasets**
>
>     | Severity          | Gaussian | Motion Blur |   Fog   |
>     |:-----------------:|:--------:|:-----------:|:-------:|
>     | Original ImageNet | 0.7984   |             |         |
>     | 1                 | 0.8158   |   0.8014     | 0.7675  |
>     | 3                 | 0.7588   |   0.7267     | 0.7220  |
>     | 5                 | 0.6364   |   0.5999     | 0.5409  |
>
>
> - **[Table 4] APGR of Model Cascading on OOD Language Datasets**
>
>     | Dataset           |  APGR   |
>     |:-----------------:|:-------:|
>     | MMLU              | 0.7013  |
>     | MMLU-Adversarial  | 0.6301  |
>
> **Observations:**
>
> - As expected, calibration and cascading gradually degrade as the degree of OOD shift increases. Under mild OOD shifts (e.g., ImageNet-C with severity levels 1 and 3, as shown in Table 1 and 3, and language-model evaluation on MMLU-Adversarial), calibrated confidence generalizes reasonably well. Under *moderate* OOD shifts (e.g., ImageNet-C with severity level 5), the method still retains a meaningful degree of generalization, though with a noticeable drop compared to the mild setting. In contrast, severe OOD (e.g., ImageNet-R) leads to a complete failure of calibration.
>
> - Interestingly, while prior work reports that smaller models are often better calibrated in-domain, we observe that after calibration, larger models (such as Eva-in22k) exhibit significantly stronger calibration robustness under OOD settings than smaller models (such as Coat_small). This is likely because Eva-in22k is pretrained on ImageNet-22K, and such large-scale pretraining makes the OOD settings less out-of-domain.

---

> ### Author Response · Authors · 2025-11-28
> **General reply (3/5)**
>
> One can introduce additional probability-density estimation to mitigate overconfidence under OOD settings. However, even generative models that explicitly estimate density still face substantial risks of misclassification in OOD detection. For instance, generative models may assign higher likelihood to OOD samples [4][7][8][9], and large language models can exhibit hallucinations [5]. Precise OOD handling requires precise generalization, which remains one of the core challenges in this area. Despite many existing improvements (such as [7][8][9]), reliable OOD detection continues to be an open problem.
>
> ### 3. Routing and other Comparisons.
> R2 and R3 suggest that the cascading method should be compared with more multi-model approaches such as routing.
>
> As suggested by the reviewers, we have added the following comparisons. For language tasks, we compare with a widely cited routing-based method Matrix Factorization[1]. For vision tasks, we compare with a MoE-based approach [3]. To ensure faithfulness and practicality under the rebuttal time constraints, we adopt the same baselines and experimental protocols as used in the original papers of these methods.
>
> For the routing baseline in language models, we make our best effort to faithfully reproduce its performance using three settings:
>
> - using the provided pretrained checkpoint (as the original work claims *“our routers have learned common characteristics of queries that allow them to distinguish between strong and weak models, generalizing to new models at inference time without additional training”* [1]);
>
> - fine-tuning the pretrained checkpoint on the target task dataset;
>
> - training the router from scratch on the target task dataset.
>
> As shown in Table [5], across all settings, the results consistently show that the proposed method significantly outperforms these baselines while maintaining its simplicity, **suggesting that the potential of simple approaches has not yet been fully explored.**
>
>
> [Table 5] APGR comparison of model cascading methods on language tasks.
> | Method                               |   MMLU   |  GSM8K  |
> | :-----------------------------------: | :------: | :-----: |
> | Random Baseline                      |  0.5000  | 0.5000  |
> | Matrix Factorization (Pretrained)     |  0.5418  | 0.5672  |
> | Matrix Factorization (Trained)        |  0.5267  | 0.6234  |
> | Matrix Factorization (Finetuned)      |  0.5497  | 0.5718  |
> | Calibrated confidence cascading (Histogram Binning, 15 bins)      |  0.7006  | 0.7489  |
> | Calibrated confidence cascading (Temperature Scaling, 15 bins)    |  **0.7013**  | **0.7593**  |
> | Calibrated confidence cascading (Platt Scaling, 15 bins)          |  **0.7013**  | 0.7496  |
>
> For the baseline in vision models, we follow the original paper and use the same DeiT model as the large model. In addition, we introduce a much smaller TinyViT model as the lightweight model for cascading. Notably, we observe that combining two models with individual accuracies in the 81–82% range produces a cascading accuracy of 90%, while simultaneously reducing the total number of activated parameters during inference by a substantial margin as shown in Table[6].
>
> [Table 6] Performance comparison of model cascading methods on language tasks.
> | Method                               | Active Param | Top-1 ACC |
> | :-----------------------------------: | :----------: | :--------: |
> | Single model (DeiT-B)      |   86.57      |   81.98    |
> | Single model (TinyViT)      |   11.00               |   81.53          |
> | MoEE (DeiT-B)       |   58.55      |   80.12    |
> | Calibrated confidence cascading (DeiT-B + TinyViT) (Temperature Scaling, 15 bins K=10) |   **39.85**   |   **90.46**   |
> | Calibrated confidence cascading (DeiT-B + TinyViT) (Histogram Binning, Training-Free, 15 bins K=10) |   **39.85**   |   **90.46**   |

---

> ### Author Response · Authors · 2025-11-28
> **General reply (4/5)**
>
> ### 4. Model Calibration Seems to Provide Marginal Improvement.
>
> Reviewers (R1, R3, R4) noted that in Figures 2 and 3, some “calibrated” and “uncalibrated” curves in the cascading experiments appear similar, suggesting that calibration may offer only marginal gains.
>
> The proposed confidence-based model cascading method consists of two parts:
>
> - **Part 1:** On the validation set of Model 1, samples are sorted by confidence and then grouped into N bins of equal size using histogram equalization.
>
> - **Part 2:** Post-hoc calibration methods (e.g., temperature scaling or Platt scaling) are applied to obtain calibrated confidence, which is subsequently used to compute the calibrated advantage within each bin.
>
> Importantly, **Part 1 already performs a non-parametric form of confidence estimation using validation data.** The histogram equalization step is closely related to histogram-based calibration [6] and therefore introduces implicit calibration. In Figures 2 and 3, “uncalibrated” and “calibrated” refer only to the confidence **before vs. after** applying the post-hoc scaling in Part 2. Thus, referring to the “uncalibrated” curves as a true uncalibrated baseline is inaccurate and potentially misleading.
>
> In scenarios where models are already well calibrated (e.g., small vision models on standard classification tasks or language models on certain NLP benchmarks), the non-parametric estimation from Part 1 alone can be sufficient. In contrast, in more poorly calibrated regimes (e.g., cascading two large vision models), the calibrated confidence derived from Part 2 becomes essential.
>
> To avoid confusion, we have added the following experiments to strengthen the evidence and will revise accordingly for better clarity:
>
> - To demonstrate the impact highlighted in Part 2, we additionally conduct calibration and cascading experiments on more severely miscalibrated data. As shown in **Table[7]**, we re-perform post-hoc calibration on ImageNet-C with severity level 5, where the raw model exhibits poor calibration. After applying temperature scaling, the ECE is reduced to a reasonable range, and the corresponding cascading results are shown in the same table. We observe that as the ECE decreases, the cascading performance consistently improves to varying degrees. This shows that when a model’s calibration error is too large to support reliable cascading decisions, post-hoc calibration becomes necessary.
>
> - As suggested by Reviewer R2 and R3, we have added a well-established routing-based method [1] and an MoE method [3] as additional baselines to provide further evidence (Part 1 + Part 2), as shown in Table 5 and Table 6. The results show that the proposed cascading approach in the manuscript significantly outperforms these baselines while maintaining its simplicity.
>
> [Table 7] ECE and APGR of model cascading on more uncalibrated settings (ImageNet-C, Severity = 5)
>
> | APGR (ECE)         | Gaussian Noise     | Motion Blur        |   Fog            |
> |---------------------|--------------------:|--------------------:|------------------:|
> | Before Calibration  | 0.6364 (0.0584)     | 0.5999 (0.1602)     | 0.5409 (0.3140)   |
> | After Calibration   | 0.6364 (0.0537)     | 0.6122 (0.0589)     | 0.6888 (0.0285)   |

---

> ### Author Response · Authors · 2025-11-28
> **General reply (5/5)**
>
> ### 5.Calibration methods.
>
> Reviewers (R1, R2, R4) question the use of task-specific calibration methods, and suggest a more thorough exploration of different calibration techniques.
>
> The proposed framework is universal and compatible with any post-hoc calibration method. We previously adopted Platt scaling for language tasks mainly for practicality, as MMLU and coding tasks provide verifiable binary rewards. In fact, any calibration technique can be applied to both vision and language models within the proposed framework.
>
> To make this point clearer, we can further include a training-free, non-parametric calibration method—histogram binning [6] — as an additional comparison. The following results evaluate three calibration methods: temperature scaling, Platt scaling, and histogram binning, in terms of both ECE and cascading performance. All three methods exhibit comparable performance, which we attribute to the favorable properties of model confidence (approximate monotonicity and generalization).
>
> [Table 8] ECE of different calibration methods.
> | ECE / Dataset         |  MMLU  | MBPP |  GSM8K  |
> | :------------------- | :----: | :-----------: | :-----: |
> | Histogram Binning    | 0.0239 |    0.0719     | 0.0536  |
> | Temperature Scaling  | 0.0768 |    0.1635     | 0.1054  |
> | Platt Scaling        | 0.0203 |    0.0696     | 0.0731  |
>
> [Table 9] APGR of model cascading of different calibration methods.
> | AGPR / Dataset        |  MMLU  |  MBPP  |  GSM8K |
> | :------------------- | :----: | :----: | :----: |
> | Histogram Binning    | 0.7006 | 0.6609 | 0.7489 |
> | Temperature Scaling  | 0.7013 | 0.6772 | 0.7593 |
> | Platt Scaling        | 0.7013 | 0.6897 | 0.7496 |
>
>
>
> [1] RouteLLM: Learning to Route LLMs with Preference Data. ICLR 2025
>
> [2] Confident Learning: Estimating Uncertainty in Dataset Labels. JAIR
>
> [3] Efficient Data Driven Mixture-of-Expert Extraction from Trained Networks. CVPR 2025
>
> [4] Do Deep Generative Models Know What They Don't Know? ICLR 2019
>
> [5] A Survey on Hallucination in Large Language Models: Principles, Taxonomy, Challenges, and Open Questions. ACM Transactions on Information Systems Jan 2024
>
> [6] Obtaining calibrated probability estimates from decision trees and naive Bayesian classifiers. ICML 2001
>
> [7] Why Normalizing Flows Fail to Detect Out-of-Distribution Data? NeurIPS 2020
>
> [8] WAIC, but Why? Generative Ensembles for Robust Anomaly Detection
>
> [9] Input complexity and out-of-distribution detection with likelihood-based generative models. ICLR 2020
>
> **Next, we reply to each reviewer to address the specific questions.**

---

### Meta-Review · Area_Chair_3h85 · 2025-12-29

**Summary:**

This submission has received diverging evaluations, with one reviewer supporting acceptance while the remaining three recommend rejection. The primary concerns raised by the reviewers are summarized below:

1. Limited novelty and contribution. All reviewers (mxpA, nfCf, xhYV, tASX) note that the core observations appear to have been previously reported, and the technical contribution and originality are not clearly demonstrated.

1. Insufficient empirical evaluation. Reviewers consistently request additional experiments, including out-of-distribution scenarios for both vision and language models, as well as stronger baselines beyond random routing (mxpA, nfCf, xhYV, tASX).

1. Marginal empirical gains. Reviewers mxpA and xhYV question whether the improvements over the uncalibrated baseline are substantial enough, and request clearer justification of the proposed method’s advantages.

1. Misleading or ambiguous claims Concerns are raised regarding the use of the term “training-free,” as well as ambiguity in the definitions of key concepts such as Mixture-of-Experts (MoE).

1. Questionable assumptions. Specifically, the assumption of independence between experts is viewed as unrealistic and insufficiently justified.

**Reviewer Concerns:**

The authors provided a detailed rebuttal addressing each reviewer concern. It appears that the last two points have been reasonably clarified. For the first three concerns:

Regarding Concern 1 (novelty and contribution), the authors reiterate that their primary contribution lies not in the observations but in the proposed confidence-based model cascading framework and the accompanying data-cleaning strategy.

For Concern 2 (empirical evaluation and baselines), the authors expanded their experiments to include additional out-of-distribution (OOD) datasets,  ImageNet-R and ImageNet-C (three corruption cases) for vision, and MMLU-Adversarial for language, to better demonstrate effectiveness in OOD scenarios. They also included a MoE-based method as a stronger baseline.

To address Concern 3 (marginal empirical improvements), the authors clarify that the perceived small performance gap stems from the fact that Part 1 of their method already incorporates a form of confidence estimation, which may reduce the observed margin. To strengthen their claim, they provide new ImageNet-C results illustrating clearer differences, and highlight improvements over both their original baseline and the newly added baseline.


In the AC’s view, the first two concerns appear to have been only partially addressed, particularly the second one. While the authors added a substantial amount of new experimental results during the rebuttal period, the evaluation still falls short of what would be considered comprehensive. For instance, the vision experiments do not cover all ImageNet-C corruption types, Stylized ImageNet, ImageNet-V2, and other common OOD benchmarks; similarly, additional language datasets would be needed to establish that the improvements are consistently significant under diverse OOD distributions. Given that the primary contribution lies in the proposed methodology, the AC feels it is premature to accept this submission without more thorough verification of its empirical performance.

**Reviewer Scores:**

Reviewer mxpA (initial score: 4) is likely to increase to a 6 based on the newly added experiments.

Reviewer nfCf (initial score: 2) may increase to a 4 due to the clarifications and additional experiments, but is unlikely to go higher given the remaining concerns about novelty and contribution.

Reviewer xhYV (initial score: 2) may increase to a 4 or remain at the initial rating. While the rebuttal provides clarification, the concerns about novelty and contribution still persist.

Reviewer tASX (initial score: 8) is likely to maintain their positive evaluation.

---

### Decision · Program_Chairs · 2026-01-26

Reject